# BANDITSPEC: Adaptive Speculative Decoding via Bandit Algorithms

**Yunlong Hou** [* † 1] **Fengzhuo Zhang** [* † ‡ 1] **Cunxiao Du** [* 2] **Xuan Zhang** [* 3] **Jiachun Pan** [1] **Tianyu Pang** [2]
**Chao Du** [2] **Vincent Y. F. Tan** [1] **Zhuoran Yang** [4]

## Abstract

Speculative decoding has emerged as a popular method to accelerate the inference of Large Language Models (LLMs) while retaining their superior text generation performance. Previous methods either adopt a fixed speculative decoding configuration regardless of the prefix tokens, or train draft models in an offline or online manner to align them with the context. This paper proposes a *training-free* online learning framework to adaptively choose the configuration of the hyperparameters for speculative decoding as text is being generated. We first formulate this hyperparameter selection problem as a Multi-Armed Bandit problem and provide a general speculative decoding framework BANDITSPEC. Furthermore, two bandit-based hyperparameter selection algorithms, UCBSPEC and EXP3SPEC, are designed and analyzed in terms of a novel quantity, the *stopping time regret*. We upper bound this regret under both stochastic and adversarial reward settings. By deriving an information-theoretic impossibility result, it is shown that the regret performance of UCBSPEC is optimal up to universal constants. Finally, extensive empirical experiments with LLaMA3 and Qwen2 demonstrate that our algorithms are effective compared to existing methods, and the throughput is close to the oracle best hyperparameter in simulated real-life LLM serving scenarios with diverse input prompts.

## 1. Introduction

A Large Language Model (LLM) is trained to predict the probability of the next token conditioned on all previous

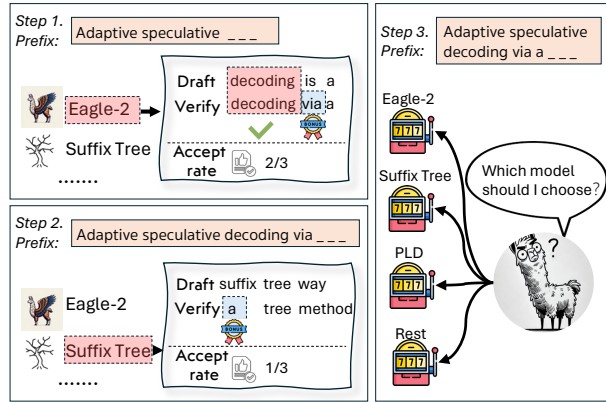

*Figure 1.* Given the prefix tokens and the candidate hyperparameter configurations (e.g., models), which configuration should be selected to decode the next tokens? We formulate this problem as a bandit problem and propose a general framework BANDITSPEC.

tokens (Brown et al., 2020; Touvron et al., 2023). This autoregressive decoding approach involves *multiple* forward passes, with each pass generating one token sequentially. Consequently, this process can lead to significant latency during inference.

Speculative decoding was introduced by Leviathan et al. (2023); Chen et al. (2023) to accelerate the inference of LLMs. The standard speculative decoding framework has been extended with improved performance since then. A thorough overview is presented at Appendix A. While the existing speculative decoding methods are diverse, most previous works adopt a *fixed* one across tasks, severely limiting their potential. For instance, when dealing with code debugging or grammar-checking tasks, the generated tokens are expected to resemble most of the input tokens. Therefore, retrieval-based speculative decoding techniques are preferred (Hu et al., 2024). In contrast, for story generation tasks, we expect the generated tokens to be more creative. Thus, a draft model with a high-temperature parameter is preferred over retrieval-based methods. The potential of these speculative decoding methods can only be exploited when the configuration of hyperparameters is *well-aligned* to the given task. There are existing works that attempt to achieve this goal, e.g., Zhou et al. (2024) distills the draft

---
*Equal contribution [†] Work done as an associate member at Sea AI Lab [‡] Project Lead [1] National University of Singapore [2] Sea AI Lab [3] Singapore Management University [4] Yale University. Correspondence to: Cunxiao Du <cnsdunm@gmail.com>, Zhuoran Yang <zhuoran.yang@yale.edu>.

*Proceedings of the 42nd International Conference on Machine Learning*, Vancouver, Canada. PMLR 267, 2025. Copyright 2025 by the author(s).

model during inference. Furthermore, even when the choice of the draft model is optimized, the associated hyperparameters can still be refined. For instance, Liu et al. (2024) and Huang et al. (2024) aim to optimize the speculation length in a training and training-free manner. Based on these observations, we ask the questions (see Figure 1) : given prefix prompts and candidate configurations of hyperparameters, is there a theoretically sound framework to model and solve the hyperparameters selection problem? Is there any *training-free* method that can *adaptively* choose the hyperparameters such that the latency of speculative decoding can be minimized?

In this paper, we answer these questions affirmatively. We adopt a bandit framework to leverage its adaptivity in unknown environments to achieve this goal. Our contributions can be summarized as follows.

• We formulate the hyperparameter selection problem in speculative decoding as a bandit problem and propose a general speculative decoding framework BANDITSPEC(ALG) (see Algorithm 3), where the hyperparameter selection algorithm ALG selects the hyperparameters to be deployed in each round of speculative decoding. The objective is to minimize the *stopping time regret*, which measures the latency of ALG compared to that of the best hyperparameter.

• Under mild stochastic and adversarial reward assumptions, we devise two hyperparameter selection algorithms, UCB-SPEC and EXP3SPEC, respectively. By deriving upper bounds on the stopping time regret, we prove that the inference latency between the proposed algorithms and the best hyperparameter under a given initial prompt vanishes asymptotically. In addition, we show, via deriving an information-theoretic impossibility result, that the regret performance of UCBSPEC is *optimal* up to constants.

• Extensive empirical experiments with LLaMA3 and Qwen2 are conducted to demonstrate the efficacy of the proposed framework. When the batch size is 1, the adaptive selection of models via UCBSPEC and EXP3SPEC can greatly improve the latency, exhibiting competitive performance against the existing methods. Under simulated real-life scenarios where LLMs are implemented for diverse prompts simultaneously, the adaptive selection of speculation length via UCBSPEC achieves comparable throughput with the oracle best.

## 2. Preliminaries

**LLM Decoding** We denote an LLM as $P : \mathcal{X}^* \to \Delta_\mathcal{X}$, where $\mathcal{X}$ and $\mathcal{X}^*$ are the space of tokens and the space of all token sequences, respectively. Most LLMs predict this conditional probability via predicting the *logits* of the next token. Concretely, the LLMs predict $\log P(x_t \mid x_{1:t-1})$, where $x_t \in \mathcal{X}$ and $x_{1:t-1} \in \mathcal{X}^*$ are

---

**Algorithm 1** CANONICAL DECODING
**Inputs:** initial prompt $\mathrm{pt}_0 = \mathrm{pt} \in \mathcal{X}^*$, target model $P$.
**Procedures:**
1: Set $t = 0$.
2: **while** $t \neq 0$ and $x_t \neq \mathrm{EOS}$ **do**
3:     $t = t + 1$.
4:     $x_t \sim P(\cdot \mid \mathrm{pt}_{t-1})$.
5:     $\mathrm{pt}_t = \mathrm{concat}(\mathrm{pt}_{t-1}, x_t)$.
6: **end while**
7: **return** $t, \mathrm{pt}_t$

---

respectively the $t$-th token and first $t-1$ tokens. In the inference stage, LLMs use an additional *temperature* parameter $\gamma > 0$ to predict the next token's probability as $\mathrm{softmax}(\gamma^{-1} \log P(\cdot \mid x_{1:t-1}))$, where $\mathrm{softmax}$ is the softmax operator. The results in our work hold for any $\gamma > 0$, and we just denote $\mathrm{softmax}(\gamma^{-1} \log P)$ as $P$ for ease of notation. When $\gamma > 0$, we sample the next token from the predicted distribution, which is called *sampling* (sampling decoding). When $\gamma \downarrow 0$, the next token will be the token that corresponds to the highest logit value; this is called *greedy decoding*. We note that the greedy decoding is deterministic, i.e., the token is sampled from a degenerate distribution. These two families of decoding methods can be unified as Algorithm 1, where LLMs autoregressively generate tokens until the EOS token.

**Speculative Decoding** As shown in Algorithm 1, the autoregressive decoding feature requires multiple forward inferences of LLMs $P$ *sequentially*. To reduce the number of forward inferences, Leviathan et al. (2023); Chen et al. (2023) proposed the vanilla speculative decoding algorithm which implements a draft model $Q$ to generate draft tokens and let the target model $P$ verify them in *parallel*. For completeness, we present and describe the vanilla speculative decoding algorithm in Appendix C.1. This vanilla specualtive decoding is then extended by some existing works, e.g., Miao et al. (2023); Cai et al. (2024) organizes the draft tokens as a tree, which improves the number of accepted tokens. The speculative decoding algorithm contains many hyperparameters, e.g., the draft model $Q$, and the tree structure in Miao et al. (2023); Cai et al. (2024). Most existing works keep these hyperparameters fixed for all the tasks. Some other works optimize the draft model in an online or offline manner (Liu et al., 2023) and the size of the tree (Chen et al., 2024), which are designed for specific considerations. In contrast, our work aims to derive a unified online hyperparameter selection algorithm that can be applied for *any* type of hyperparameters.

**Multi-Armed Bandits** The Multi-Armed Bandit (MAB) is a fundamental online decision-making problem (see Algorithm 7 for its dynamics). In its classical **stochastic** form, an agent chooses from $K$ arms, each of which delivers a re-

**Algorithm 2** SPECUATIVE DECODING SUBROUTINE (SPECDECSUB)

**Inputs:** pt $\in \mathcal{X}^*$, target model $P$, the hyperparameters $S$, maximum speculation length $L$.
**Procedures:**
1: Call a standard speculative decoding algorithm with $(\mathrm{pt}, P, S, L)$.
2: **return** the accepted and bonus tokens $x_{1:\tau}$, where $\tau \geq 1$.

---

**Algorithm 3** SPECULATIVE DECODING WITH BANDITS (BANDITSPEC)

**Inputs:** arm selection algorithm ALG, initial prompt $\mathrm{pt}_0 = \mathrm{pt} \in \mathcal{X}^*$, bandit configuration $\nu = (P, \mathcal{S} = \{S_i\}_{i \in [K]}, L)$.
**Procedures:**
1: $t = 0, \mathcal{H}_0 = \emptyset, I_0 = 1, x_{I_0,0} = \emptyset$.
2: **while** EOS $\notin x_{I_t,t}$ **do**
3:    $t = t + 1$.
4:    Select a hyperparameter index $I_t = \mathrm{ALG}(\mathcal{H}_{t-1})$.
5:    $x_{I_t,t} = \mathrm{SPECDECSUB}(\mathrm{pt}_{t-1}, P, S_{I_t}, L)$.
6:    $\mathrm{pt}_t = \mathrm{concat}(\mathrm{pt}_{t-1}, x_{I_t,t})$.
7:    $\mathcal{H}_t = \mathrm{concat}(\mathcal{H}_{t-1}, (I_t, x_{I_t,t}))$.
8: **end while**
9: **return** $\mathrm{ST}(\mathrm{ALG}, \mathrm{pt}, \nu) = t$, $\mathrm{pt}_{\mathrm{ST}(\mathrm{ALG},\mathrm{pt},\nu)} = \mathrm{pt}_t$.

---

ward sampled i.i.d. from an unknown but fixed distribution when pulled (Lattimore & Szepesvári, 2020). The goal is to select arms over $T$ rounds to maximize cumulative rewards. Two primary classes of algorithms—UCB-type (Auer et al., 2002b) and sampling-based methods (Russo et al., 2017)—have been developed and proven optimal in this setting. In the **adversarial** formulation, there are no assumptions on the reward distributions; rewards can evolve arbitrarily over time and may be correlated across arms (Auer et al., 2002a). Several algorithms, such as EXP3 and EXP4 (Auer et al., 2002a), are known to achieve optimal performance under these conditions. In this work, we frame the hyperparameter selection problem as an MAB problem and develop algorithms tailored to both stochastic and adversarial settings.

**Notations:** Let $[N] := \{1, \cdots, N\}$. For a finite set $\mathcal{X}$, we denote the set of distributions supported on it as $\boldsymbol{\Delta}_{\mathcal{X}} = \{P : \mathcal{X} \to [0,1] \mid \sum_{x \in \mathcal{X}} P(x) = 1, P(x) \geq 0 \text{ for all } x \in \mathcal{X}\}$. The space of all finite length sequences whose components belong to $\mathcal{X}$ is denoted as $\mathcal{X}^*$, and we use $x_{1:L} \in \mathcal{X}^L \subseteq \mathcal{X}^*$ to denote a length-$L$ sequence. The Kullback–Leibler (KL) divergence between two distributions $P$ and $Q$ is denoted as $\mathrm{KL}(P, Q)$.

## 3. Bandits for Adaptive Speculative Decoding

In the section, we formally formulate the *hyperparameter selection problem* in speculative decoding using the parlance of multi-armed bandits. The goal of this online decision-making process is to decode as soon as possible, i.e., minimizing the *latency* of the LLM decoding. Different from the classical multi-armed bandit problem, this problem involves two stochastic processes that march at various paces. In fact, as described in Appendix C.1, each (vanilla) speculative decoding subroutine produces several tokens, where the number of accepted tokens itself is also a random variable. Thus, the selection of hyperparameters of each speculative decoding subroutine and the token generation processes are evolving at different paces.

To put the problem in a mathematically sound way, we first specify a general speculative decoding subroutine (SPECDECSUB) in Algorithm 2. The input of this subroutine is a prompt pt $\in \mathcal{X}^*$, a target model $P$, a specification of hyperparameters $S$, and the maximum speculation

length $L$, and the output is the accepted token sequence $x_{1:\tau} \in \mathcal{X}^*$. We provide two examples of the hyperparameter sets here. (1) If we adopt the vanilla speculative decoding (Algorithm 6) as Line 1, $S$ can be different draft models $Q : \mathcal{X}^* \to \boldsymbol{\Delta}_{\mathcal{X}}$, and $\mathcal{S}$ is the set of all the provided draft models. We would like to choose a draft model according to its training context, e.g. math, creative writing, to decode the current prefix. Then the problem we consider is how to adaptively select a proper draft model for speculative decoding via bandit algorithms. (2) If we adopt Medusa (Cai et al., 2024) as Line 1, $S$ can be different tree structures, and $\mathcal{S}$ is the set of plausible tree structures. In this problem, we would like to adaptively adjust the speculation tree structure according to the context.

With the help of SPECDECSUB, the speculative decoding with a bandit framework, BANDITSPEC, can be specified in Algorithm 3 and as illustrated in Figure 1. The *bandit configuration* $\nu = (P, \mathcal{S} = \{S_i\}_{i \in [K]}, L)$ consists of three components: the target model $P$, the set of $K$ candidate hyperparameter specifications $\mathcal{S}$, and the maximum speculation length $L$. Each hyperparameter specification $S_i \in \mathcal{S}$ corresponds to an arm in the bandit problem. Given a prompt pt and an arm selection algorithm ALG, a hyperparameter specification is chosen according to the history $\mathcal{H}_{t-1}$ in Line 4. Then SPECDECSUB is invoked with selected hyperparameters $S_{I_t}$ as input. The output of SPECDECSUB, $x_{I_t,t}$,[1] is then adopted to update the prompt (Line 6) and the history information (Line 7). The whole process stops when the EOS token appears in the prompt. We denote the number of calls to SPECDECSUB (the stopping time) and the generated token sequence as $\mathrm{ST}(\mathrm{ALG}, \mathrm{pt}, \nu)$ and $\mathrm{pt}_{\mathrm{ST}(\mathrm{ALG},\mathrm{pt},\nu)}$, respectively. To minimize the decoding latency, we aim to design ALG to minimize $\mathrm{ST}(\mathrm{ALG}, \mathrm{pt}, \nu)$. Since the position of the EOS token itself is a random variable, we would like

---

[1] We abbreviate the notation $x_{I_t,t,1:\tau_t}$ as $x_{I_t,t}$ to represent the accepted tokens generated by $S_{I_t}$ at time step $t$.

to minimize the expectation of $\mathrm{ST}(\mathtt{ALG}, \mathrm{pt}, \nu)$. The performance of $\mathtt{ALG}$ is measured via the *stopping time regret*

$$\mathrm{Reg}(\mathtt{ALG}, \mathrm{pt}, \nu) := \mathbb{E}\big[\mathrm{ST}(\mathtt{ALG}, \mathrm{pt}, \nu) \mid \mathrm{pt}, \nu\big] \qquad (1)$$
$$- \mathbb{E}\big[\mathrm{ST}(\mathtt{ALG}_{i^*(\mathrm{pt}, \nu)}, \mathrm{pt}, \nu) \mid \mathrm{pt}, \nu\big],$$

where $\mathtt{ALG}_i$ is the arm selection algorithm which adopts $S_i$ in all rounds, i.e., $i = \mathtt{ALG}_i(\mathcal{H}_t)$ for all $\mathcal{H}_t$ and $t$, and $i^*(\mathrm{pt}, \nu) = \mathrm{argmin}_{i \in [K]} \mathbb{E}\left[\mathrm{ST}(\mathtt{ALG}_i, \mathrm{pt}, \nu) \mid \mathrm{pt}, \nu\right]$ denotes the index of the best hyperparameter for prompt $\mathrm{pt}$ under bandit configuration $\nu$.

For ease of notation, when $\nu$ and $\mathrm{pt}$ are clear from the context, $\mathrm{ST}(\mathtt{ALG}, \mathrm{pt}, \nu), \mathrm{Reg}(\mathtt{ALG}, \mathrm{pt}, \nu)$, and $i^*(\mathrm{pt}, \nu)$ will be abbreviated as $\mathrm{ST}(\mathtt{ALG}), \mathrm{Reg}(\mathtt{ALG})$, and $i^*$, respectively. We will use $\mathrm{BANDITSPEC}(\mathtt{ALG})$ to specify the choice of $\mathtt{ALG}$ in Algorithm 3. For simplicity, we regard the bonus token as the last accepted token. Thus, the length of the accepted tokens $x_{I_t,t}$ at each round, $y_{I_t,t}$, is between $[1, L+1]$.

Before going to the algorithm design and the theoretical analysis, we would like to specify some important properties that are shared for any arm selection algorithm $\mathtt{ALG}$ and clarify the intuitions about our theoretical analysis. We denote the stopping time of the canonical decoding (Algorithm 1) and the generated sequence as $\tau_c$ and $\mathrm{pt}_{\tau_c}$, respectively.

**Proposition 3.1.** *For any arm selection algorithm* $\mathtt{ALG}$ *that selects an arm according to the history, the generated prompt* $\mathrm{pt}_{\mathrm{ST}(\mathtt{ALG})}$ *is equal to* $\mathrm{pt}_{\tau_c}$ *in distribution, i.e.,*

$$\mathrm{pt}_{\mathrm{ST}(\mathtt{ALG})} \overset{d}{=} \mathrm{pt}_{\tau_c}, \text{ and } \mathrm{len}(\mathrm{pt}_{\mathrm{ST}(\mathtt{ALG})}) \overset{d}{=} \mathrm{len}(\mathrm{pt}_{\tau_c}). \quad (2)$$

*The stopping time* $\mathrm{ST}(\mathtt{ALG})$ *can be bounded as*

$$\frac{\mathrm{len}(\mathrm{pt}_{\mathrm{ST}(\mathtt{ALG})})}{L + 1} \leq \mathrm{ST}(\mathtt{ALG}) \leq \mathrm{len}(\mathrm{pt}_{\mathrm{ST}(\mathtt{ALG})}), \ a.s. \quad (3)$$

The proof of Proposition 3.1 is provided in Appendix D.1. This proposition states that the distribution of the generated prompt is the same as that of the prompt generated by Algorithm 1. The stopping time $\mathrm{ST}(\mathtt{ALG})$ is equal to the length of the generated prompt up to a constant. To facilitate our theoretical understanding, we pose the following question.

**Question:** *Whether it is possible to devise an arm selection algorithm* $\mathtt{ALG}$ *to achieve* **sublinear** *regret in terms of the length of the generated token sequence, i.e., is* $\mathrm{Reg}(\mathtt{ALG}, \mathrm{pt}, \nu) = o(\mathbb{E}[\mathrm{len}(\mathrm{pt}_{\mathrm{ST}(\mathtt{ALG})})])$?

**Interpretation of the Desired Result.** Given a prompt $\mathrm{pt}$ and bandit configuration $\nu$, BANDITSPEC adpatively selects the hyperparameter via $\mathtt{ALG}$ and learns the context. The stopping time regret (1) measures how the stopping time of $\mathrm{BANDITSPEC}(\mathtt{ALG})$ compared to that of the (agnostic) best one $\mathrm{BANDITSPEC}(\mathtt{ALG}_{i^*})$. By minimizing

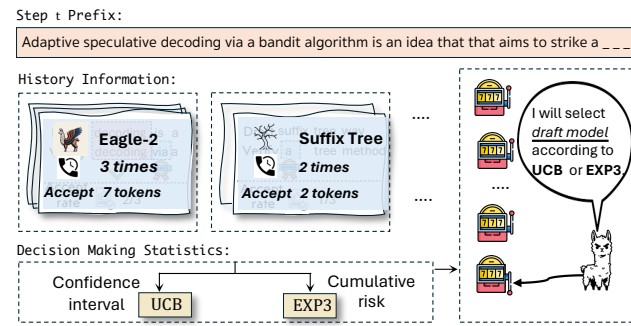

*Figure 2.* Illustration of our bandit model for choosing configurations to decode the next token, where UCB and EXP3 refer to UCBSPEC and EXP3SPEC, repectively.

$\mathrm{Reg}(\mathtt{ALG}, \mathrm{pt}, \nu)$, we want to devise an $\mathtt{ALG}$ to (approximately) match the performance of $\mathtt{ALG}_{i^*}$. In particular, if $\mathrm{Reg}(\mathtt{ALG}, \mathrm{pt}, \nu) = o(\mathbb{E}[\mathrm{len}(\mathrm{pt}_{\mathrm{ST}(\mathtt{ALG})})])$, it implies that $\mathrm{BANDITSPEC}(\mathtt{ALG})$ requires the same number of speculative decoding rounds as $\mathrm{BANDITSPEC}(\mathtt{ALG}_{i^*})$ asymptotically even though the information about $S_{i^*}$ is not revealed at the beginning. In order words, $\mathrm{BANDITSPEC}(\mathtt{ALG})$ learns the identity of $S_{i^*}$ quickly and the price for this learning process can be amortized over time. Additionally, when $\mathrm{BANDITSPEC}(\mathtt{ALG})$ is deployed over diverse prompt inputs, we expect a significant acceleration of token generation compared to any fixed single speculative decoding method.

**Why do we consider stochastic and adversarial settings?** To derive efficient algorithms and meaningful theoretical analysis, it is necessary to make certain plausible assumptions of the problem. For the BANDITSPEC problem, we need to model the *stochasticity* of the number of accepted tokens for each hyperparameter specification. We highlight that in real-world applications, they are far from identically and independently distributed. The stochastic case (Section 4) models it as random variables and only assumes that each hyperparameter will have a *stationary* mean acceptance length (Assumption 4.1) without the independence assumption. The adversarial case (Section 5) removes this stationarity assumption and does not make any distributional assumption of the number of accepted tokens for each hyperparameter. We highlight that there is no explicit *adversary* in the speculative decoding, but we model the stochasticity of the number of accepted tokens as the randomness from an (imaginary) adversary.

## 4. Modeling Tokens Stochastically

In this section, we model the length of the accepted tokens as random variables.

**Assumption 4.1** (Stationary Mean Values). There exist $K$ values $\{\mu_i\}_{i \in [K]} \subset [1, L+1]$, such that conditioned on the

**Algorithm 4** UCBSPEC

**Inputs:** number of hyperparameter specifications $K$, history $\mathcal{H}_t = \left((I_s, X_{I_s,s})\right)_{s=1}^{t}$, confidence parameter $\delta$.

**Procedures:**

1: **if** $t \leq K - 1$ **then return** $I_{t+1} = t + 1$.
2: Compute the lengths $Y_{I_s,s} = \text{len}(X_{I_s,s})$ for all $s \in [t]$.
3: Set the statistics $\{\hat{\mu}_{i,t}\}_{i\in[K]}$, $\{\text{UCB}_{i,t}\}_{i\in[K]}$, where

$$n_{i,t} = \sum_{s=1}^{t} \mathbb{1}\{I_s = i\}, \ \hat{\mu}_{i,t} = \frac{\sum_{s=1}^{t} Y_{i,s}\mathbb{1}\{I_s = i\}}{n_{i,t}},$$

$$\text{cr}_{i,t} = \frac{L}{2}\sqrt{\frac{1 + n_{i,t}}{n_{i,t}^2}\left(1 + 2\log\frac{Kt^2(1+n_{i,t})^{\frac{1}{2}}}{\delta}\right)},$$

$$\text{UCB}_{i,t} = \hat{\mu}_{i,t} + \text{cr}_{i,t}.$$

4: **return** index $I_{t+1} = \text{argmax}_{i\in[K]} \text{UCB}_{i,t}$.

history $\mathcal{H}_{t-1}$ and the chosen arm $I_t$ at time $t$, the expected number of the accepted tokens $\mathbb{E}[Y_{I_t,t} \mid \mathcal{H}_{t-1}, I_t] = \mu_{I_t}$.

This assumption assumes that the conditional expectation of the number of accepted tokens for each hyperparameter is equal to a fixed number conditioned on the previous tokens. We emphasize that this assumption does *not* require independence between the number of accepted tokens across implementations of SPECDECSUB, which would be unrealistic in real-world applications. More discussions are provided in Appendix B.1.

### 4.1. Upper Bounds for the Stochastic Case

**Algorithm Design** We design a UCB-type arm selection algorithm UCBSPEC, as shown in Algorithm 4. To avoid additional terms, we call the aggregated algorithm, BANDITSPEC(UCBSPEC), as UCBSPEC. The full version of UCBSPEC is detailed in Algorithm 8.

This aggregated algorithm, UCBSPEC, is adapted from the classical UCB-1 algorithm in Auer et al. (2002b). The main differences are the confidence radius design $\text{cr}_{i,t}$ and the stopping rule. We highlight that the form of $\text{cr}_{i,t}$ is designed to fit the weak assumption of the number of accepted tokens. In fact, the proof of the regret of UCB-1 assumes that the values of each arm are generated *before* the pull of arms (Auer et al., 2002b; Lattimore & Szepesvári, 2020), which bifurcates from practical LLM inference scenarios. In contrast, we remove this strong restriction. The stopping rule of UCBSPEC makes the analysis of our algorithm rather different from that of UCB-1. The stochasticity of the total number of arm pulls requires a novel regret decomposition analysis that is not presented in previous works.

**Theoretical Analysis** We first state an assumption.

**Assumption 4.2** (Finite Generation Length). *Given any*

prompt $\text{pt} \in \mathcal{X}^*$, the expected length of the output sequence of the canonical decoding algorithm (Algorithm 1) is finite, i.e., $\mathbb{E}[\text{len}(\text{pt}_{\tau_c})] < \infty$.

This assumption states that the expected length of the generated prompt is finite. In real-world applications, the length of the generated prompt is always finite due to the limits of computation and storage.

To state our main result, we denote the *suboptimality gap* between the best arm $i^* := \text{argmax}_{i\in[K]} \mu_i$ and arm $i$ as $\Delta_i := \mu_{i^*} - \mu_i$. Define the *hardness parameter* $\text{H}(\text{pt}, \nu) := \sum_{i \neq i^*} 1/(\mu_{i^*}\Delta_i)$, which captures the difficulty of acceleration given the initial prompt $\text{pt}$ and bandit configuration $\nu$.

**Theorem 4.3** (Upper Bound). *Under Assumptions 4.1 and 4.2, given any prompt $\text{pt} \in \mathcal{X}^*$ and bandit configuration $\nu = (P, \mathcal{S} = \{S_i\}_{i\in[K]}, L)$, the expected stopping time regret of Algorithm 3 with* ALG $=$ *Algorithm 4 (*UCBSPEC*) is upper bounded as*

$$\text{Reg}(\text{ALG}, \text{pt}, \nu) = O\left(\text{H}(\text{pt}, \nu) \cdot L^2 \log \mathbb{E}[\text{len}(\text{pt}_{\tau_c})]\right).$$

Theorem 4.3 answers the proposed question in Section 3 in the affirmative under Assumptions 4.1 and 4.2. To interpret the results of the theorem, for each hyperparameter $S_i$, it requires $n_i = O(L^2 \log \mathbb{E}[\text{len}(\text{pt}_{\text{ST}(\text{ALG})})]/\Delta_i^2))$ pulls to identify that $S_i$ is suboptimal under the current prompt $\text{pt}$ and bandit configuration $\nu$, resulting in $n_i\Delta_i$ token loss compared to the case where $S_{i^*}$ had been adopted. Additionally, this loss could be compensated by $n_i\Delta_i/\mu_{i^*}$ pulls of $S_{i^*}$, which constitutes the final stopping time regret bound. The proof is postponed to Appendix D.2 with more discussions in Appendix B.2.

### 4.2. Lower Bound for the Stochastic Case

We further provide an information-theoretic lower bound of the regret under the **greedy decoding** strategy to indicate how the upper bound is in Theorem 4.3. More details and the proof of Theorem 4.4 are deferred to Appendix D.4.

**Theorem 4.4** (Lower Bound). *Given any sequence of initial prompts $(\text{pt}^m)_{m=1}^{\infty} \subset \mathcal{X}_{\text{init}}^*$ with $\text{len}(\text{pt}_{\tau_c}^m) \to \infty, m \to \infty$ and a bandit configuration $\nu = (P, \mathcal{S} = \{S_i\}_{i\in[K]}, L)$, under Assumption D.4, the greedy decoding strategy and the dynamics represented in Algorithm 3, for any non-anticipatory and consistent arm selection algorithm* ALG, *the expected regret satisfies*

$$\liminf_{m\to\infty} \frac{\text{Reg}(\text{ALG}, \text{pt}, \nu)}{\log(\text{len}(\text{pt}_{\tau_c}^m))} \geq \sum_{i\neq i^*} \frac{\Delta_i}{\mu_{i^*}} \cdot \frac{1}{\text{kl}_i},$$

*where* $\text{kl}_i := \inf_{S\in\mathcal{S}}\{\text{KL}(P_{S_i}, P_S) : \mathbb{E}_{X\sim P_S}[X] > \mu_{i^*}\}$.

To provide a more concrete example of the lower bound, consider the *truncated geometric distribution* (TGD) on $[1, L+1]$ with parameter $p \in (0, 1)$, i.e.,

$$P_S(x) = \begin{cases} p^{x-1}(1-p), & x = 1, 2, \ldots, L, \\ p^L, & x = L+1. \end{cases} \quad (4)$$

This TGD was considered in the seminal works on speculative decoding (Leviathan et al., 2023; Chen et al., 2023).

**Proposition 4.5** (Tightness Result). *Let $\mathcal{S}_{\text{TGD}} = \{S : P_S \text{ satisfies } (4)\}$. Let $\{S_i\}_{i=1}^K \subset \mathcal{S}_{\text{TGD}}$ and $S_i$ satisfies (4) with $p_i$ (Line 5 in Algorithm 2), then*

$$\liminf_{m \to \infty} \frac{\text{Reg}(\texttt{ALG}, \text{pt}^m, \nu)}{\log(\text{len}(\text{pt}_{\tau_c}^m))} \geq \text{H}(\text{pt}, \nu) \cdot \frac{p_{i^*}(1 - p_{i^*}^L)}{(1 - p_{i^*})}.$$

*Therefore, the upper and lower bound match up absolute constants and a $\frac{L^2(1-p_{i^*})}{p_{i^*}(1-p_{i^*}^L)}$ factor. In particular, if $p_{i^*} \in (2^{-1/L}, 1)$, they match up to absolute constants and L.*

The proof is deferred to Appendix E.3. Proposition 4.5 indicates UCBSPEC is *optimal* up to constants and $L$ when considering the TGD. In other words, the additional speculative decoding rounds of UCBSPEC not only achieves $O(\log \mathbb{E}[\text{len}(\text{pt}_{\tau_c})])$ compared to $\texttt{ALG}_{i^*}$, but is also among the best possible for any arm selection algorithm (up to constants).

For the tightness of our algorithm, according to Note 15.3 in Lattimore & Szepesvári (2020), $\text{kl}_i = O(\Delta_i^2)$ when $\Delta_i$ is small. This indicates the dominating terms in the upper bound in Theorem 4.3 match the lower bound in Theorem 4.4 up to (possibly instance-dependent) constants. Futherfmore, because the truncated geometric distribution is more close to a sub-exponential family distribution, especially when $L$ is large, bandit algorithms built upon UCB1 (Auer et al., 2002b) are generally loose in some factors. In order to close the gap between the upper and lower bounds completely, KL-UCB (Garivier & Cappé, 2011) can possibly be adapted to this problem out of theoretical interest. However, on the practical side, KL-UCB demands solving an optimization problem at each round, which can be time-consuming during implementations. Thus, it does not perfectly align with our ultimate goal of LLM inference acceleration.

## 5. Modeling Tokens Adversarially

In this section, we weaken Assumption 4.1 in Section 4 and consider a more general case. Specifically, we make the following assumption on the number of accepted tokens.

**Assumption 5.1** (Adversarial Mean Values). Let the number of accepted tokens generated by hyperparameter $S_i$ at

---

**Algorithm 5** EXP3SPEC

**Inputs:** number of hyperparameter specifications $K$, history $\mathcal{H}_t = \left((I_s, X_{I_s,s})\right)_{s=1}^t$.

**Procedures:**
1: Compute the lengths $Y_{I_s,s} = \text{len}(X_{I_s,s})$ for all $s \in [t]$.

2: Set the statistics for all $i \in [K]$

$$\widehat{Z}_{i,t} = \mathbb{1}\{i = I_t\} \cdot \frac{L + 1 - Y_{i,t}}{L \cdot p_{t,i}}. \quad (6)$$

3: Set learning rate $\eta_t = \sqrt{\log K/(t \cdot K)}$.
4: Set probability vector $p_t \in \Delta_{[K]}$ with for all $i \in [K]$

$$p_{t,i} = \frac{\exp\left(-\eta_t \sum_{s=1}^{t-1} \widehat{Z}_{i,s}\right)}{\sum_{j=1}^K \exp\left(-\eta_t \sum_{s=1}^{t-1} \widehat{Z}_{j,s}\right)}.$$

5: **return** hyperparameter index $I_{t+1} \sim p_t$.

---

time step $t$ be $y_{i,t} = \text{len}(X_{i,t})$. We assume $\{y_{i,t}\}_{i \in [K], t \in \mathbb{N}}$ is fixed by the environment before the algorithm starts.

The bandits problem with this assumption is often referred to as the *oblivious adversarial bandits* in the online learning works (Auer et al., 2002a; Lattimore & Szepesvári, 2020). It admits more general and practical setups compared to the stochastic MAB. We find the greedy decoding strategy aligns more closely to this setup in the sense that the generated tokens by the models are (potentially) fixed given the initial prompt. Hence, we present our result under the **greedy decoding** strategy in this section.[2] Given a prompt $\text{pt} \in \mathcal{X}^*$ and a bandits configuration $\nu$, the stopping time regret (1) of an arm selection algorithm $\texttt{ALG}$ becomes

$$\text{Reg}(\texttt{ALG}) := \mathbb{E}[\text{ST}(\texttt{ALG})] - \min_{i \in [K]} \text{ST}(\texttt{ALG}_i) \quad (5)$$

It is worth pointing out that under the greedy decoding strategy, the stopping time of any proposed algorithm can still be random due to the internal randomness embedded in the algorithm. For instance, the choice of hyperparameter $S_{I_t}$ in Line 5 in Algorithm 5.

**Algorithm Design** We present our arm selection algorithm in Algorithm 5, which is an abridged version of the full version BANDITSPEC(EXP3SPEC) delineated in Algorithm 9. This algorithm modifies the anytime EXP3 algorithm (Lattimore & Szepesvári, 2020) to suit the speculative decoding application. In terms of the algorithm design, the main difference lies in the change of the stopping rule. We highlight that while the stopping time of the algorithm is random,

---

[2]Our analysis can also be extended to cover the sampling decoding strategy (see Remark D.3).

the anytime feature of Algorithm 5 does not require any information about the time horizon. This is achieved by the vanishing sequence of learning rates $\{\eta_t\}_{t \in \mathbb{N}}$ which can be elegantly adapted to the unknown stopping time. With regard to the analysis, previous works (Auer et al., 2002a; Bubeck et al., 2012; Lattimore & Szepesvári, 2020) only consider the gap between the cumulated rewards over the *same* fixed horizon $T$, i.e., $\max_{i \in [K]} \mathbb{E}[\sum_{t \in [T]} y_{i,t} - y_{I_t,t}]$. In contrast, we need to upper bound the stopping time regret in (5) where the baseline $\mathtt{ALG}_i$ and any proposed algorithm $\mathtt{ALG}$ have *different* termination times in general. Thus, the analysis is much more involved.

**Theoretical Analysis** To ease the analysis, we make an assumption on the stopping time of Algorithm 5.

**Assumption 5.2** (Stopping Time assumption)**.** Given a prompt $\mathrm{pt} \in \mathcal{X}^*$ and configuration $\nu$, let $i^* := \mathrm{argmin}_{i \in [K]} \mathrm{ST}(\mathtt{ALG}_i)$. We assume that $\mathrm{ST}(\mathtt{ALG}) > \mathrm{ST}(\mathtt{ALG}_{i^*})$ almost surely.

In speculative decoding, when the initial prompt is given, there generally exists a hyperparameter that has the highest acceptance rate in most rounds compared to the rest of the hyperparameters. As bandit algorithms will explore those suboptimal hyperparameters, the termination time falls behind that of the optimal hyperparameter. Therefore, Assumption 5.2 is satisfied in practical applications.

**Theorem 5.3.** *Under Assumptions 4.2, 5.1 and 5.2, given any prompt* $\mathrm{pt} \in \mathcal{X}^*$ *and bandit configuration* $\nu = (P, \mathcal{S} = \{S_i\}_{i \in [K]}, L)$, *the expected stopping time regret of Algorithm 3 with* $\mathtt{ALG} =$ *Algorithm 5 (*EXP3SPEC*),*

$$\mathrm{Reg}(\mathtt{ALG}, \mathrm{pt}, \nu) \le 2L \cdot \min \left\{ \sqrt{\mathrm{len}(\mathrm{pt}_{\tau_c}) K \log K}, \right.$$
$$\left. 2LK \log K + \sqrt{\min_{i \in [K]} \mathrm{ST}(\mathtt{ALG}_i) K \log K} \right\}.$$

Theorem 5.3 also provides an affirmative answer to the question posed in Section 3. The **first** term in the minimum provides a worst-case guarantee. Even if all hyperparameters in $\mathcal{S}$ are not good or $K$ is large, EXP3SPEC will stop at no more than $O(\sqrt{\mathrm{len}(\mathrm{pt}_{\tau_c})})$ time steps after $S_{i^*}$ terminates. The **second** term is an instance-dependent bound in terms of hyperparameters $\mathcal{S}$. Specifically, when the best hyperparameter $S_{i^*}$ has small stopping time, EXP3SPEC will scale as $\mathrm{ST}(\mathtt{ALG}_{i^*}) + O(\sqrt{\mathrm{ST}(\mathtt{ALG}_{i^*})})$. This upper bound suggests that the number of speculative decoding rounds of EXP3SPEC is almost the same as that of the best hyperparameter configuration $\mathtt{ALG}_{i^*}$.

# 6. Experiments

In this section, we conduct two sets of experiments to demonstrate the efficacy of the proposed bandit framework

BANDITSPEC, along with UCBSPEC and EXP3SPEC. In the first experiment, the candidate hyperparameters are different draft models. In the second experiment, the candidate hyperparameters are different speculation lengths, where real-life LLM serving scenarios are simulated with diverse input prompts. Additional experimental results on memory utilization and additional experiments on larger models and different hardwares are provided in Appendix G. The code is accessible via https://github.com/sail-sg/BanditSpec.

### 6.1. Experiment with Draft Models

**Experimental Setups** We adopt the open-sourced LLaMA3-8B-Instruct (Dubey et al., 2024) and Qwen2-7B-Instruct (Yang et al., 2024) as the target models. The commonly-used existing speculative decoding methods PLD (Saxena, 2023), Rest (He et al., 2024), Suffix Tree (Oliaro et al., 2024; Hu et al., 2024) and Eagle-2 (Li et al., 2024b) are adopted as the baselines. Among these baselines, PLD, Rest, and Suffix Tree represent the non-parametric (or model-free) speculative decoding methods, whereas Eagle-2 represents the speculative decoding methods that utilize smaller draft models. Each of these methods corresponds to an arm in our problem.

The experiments are carried out on Spec Bench (Xia et al., 2024), Alpaca (Taori et al., 2023), Code Editor (Guo et al., 2024) and Debug Bench (Tian et al., 2024). Among these benchmarks, Spec Bench and Alpaca encompass multiple topics, while Code Editor and Debug Bench focus on coding tasks, a representative scenario for specialized models.

We record the number of accepted tokens for each speculative decoding step, as well as the wall-time for generating each complete response. The Mean Accepted Tokens (MAT) and the throughput (Tokens/s) are computed. These two metrics are widely adopted in the speculative decoding community and are positively correlated (Xia et al., 2024). In particular, Tokens/s measures the actual latency during decoding. The experiments are conducted on a single A100 and set batch size as 1.

**Experimental Results** We report the results of our experiments in Table 1. The proposed adaptive speculative decoding framework BANDITSPEC exhibits superior performance compared to existing methods in the datasets we consider. In particular, the best performance measured by Token/s is always achieved by the proposed framework. We note that although the non-parametric methods are worse than Eagle-2 *in average*, they are effective on a portion of prompts. Our proposed methods, UCBSPEC and EXP3SPEC, automatically adapt to different prompts, i.e., suffering from a small stopping time regret on each prompt. Thus, they achieve better performance than all the methods that only use a fixed model. On Debug Bench, UCBSPEC can even achieve

*Table 1.* Empirical Comparison between the proposed algorithms and the existing works, measured by Mean Accepted Tokens (MAT) (↑) and Tokens/s (↑). The best result is highlighted in **bold**, while the second best result is underlined. The proposed algorithms demonstrate unequivocal superior performance compared with the existing methods.

| Methods | Spec Bench | | Alpaca | | Code Editor | | Debug Bench | |
|---|---|---|---|---|---|---|---|---|
| | MAT(↑) | Tokens/s(↑) | MAT(↑) | Tokens/s(↑) | MAT(↑) | Tokens/s(↑) | MAT(↑) | Tokens/s(↑) |
| ***LLaMA3-8B-Instruct*** | | | | | | | | |
| Vanilla | 1.00 | 35.73 | 1.00 | 35.92 | 1.00 | 36.32 | 1.00 | 36.89 |
| PLD | 1.46 | 43.96 | 1.53 | 53.06 | 2.13 | 82.61 | 1.67 | 82.76 |
| Rest | 1.29 | 40.67 | 1.48 | 52.40 | 1.33 | 51.32 | 1.29 | 48.49 |
| Suffix Tree | 1.83 | 55.10 | 1.71 | 64.02 | 2.30 | 90.21 | 2.13 | 77.56 |
| Eagle-2 | 3.94 | 98.15 | 4.04 | 110.00 | 4.79 | 128.76 | **4.78** | 119.12 |
| EXP3SPEC | 3.65 | 102.10 | 4.23 | 120.38 | 4.36 | 137.29 | 4.50 | 132.25 |
| UCBSPEC | **3.98** | **105.72** | **4.35** | **125.78** | **4.83** | **138.27** | 4.60 | **135.34** |
| ***Qwen2-7B-Instruct*** | | | | | | | | |
| Vanilla | 1.00 | 38.71 | 1.00 | 39.32 | 1.00 | 39.30 | 1.00 | 39.57 |
| PLD | 1.55 | 52.44 | 1.42 | 58.41 | 1.89 | 64.56 | 2.15 | 70.49 |
| Rest | 1.31 | 46.42 | 1.47 | 59.01 | 1.31 | 53.79 | 1.22 | 50.51 |
| Suffix Tree | 1.96 | 68.42 | 1.46 | 62.60 | 2.18 | 85.75 | 2.49 | 101.47 |
| Eagle-2 | 3.64 | 97.82 | 3.61 | 104.43 | 4.88 | 138.58 | 4.79 | 126.01 |
| EXP3SPEC | 3.76 | 107.36 | 3.83 | 113.90 | 4.90 | 160.41 | 4.86 | **151.73** |
| UCBSPEC | **4.13** | **112.33** | **3.93** | **114.20** | **4.92** | **161.35** | **5.10** | 151.37 |

improvements of 13% for LLaMA3 and 19% for Qwen2. Moreover, as UCBSPEC demonstrates better performance under almost all benchmarks with both two target models, this suggests that speculative decoding in real-life environments tends to be closer to the stochastic (Assumption 4.1) compared to the adversarial reward case (Assumption 5.1).

*Remark* 6.1. The adversarial setting can be regarded as a means of comparison to the stationary setting. Prior to this work, it was *a priori* unclear how to use MAB to improve speculative decoding. Should one employ a stochastic, adversarial or even more generalized model? We consider a range of such MAB models and do a comparison among them to provide the community with a guide on which MAB model is best suited to the speculative decoding problem. As the empirical performance of UCBSPEC is better than EXP3SPEC (Table 1), it implies that real-life scenario tends to be benign and may be more aligned with the stationary mean assumption.

### 6.2. Experiment with Speculation Lengths

**Experimental Setups** In addition to improving the latency when batch size is 1, our proposed algorithms also improve the throughput in real LLM serving scenarios with different batch sizes. In practical serving environments, speculative decoding does not always yield performance gains due to variations in batch size and acceptance rate. As the batch size increases, the system rapidly becomes compute-bound,

while a lower acceptance rate can lead to wasted computation resources of GPU. Additionally, the execution time of the draft model contributes to an overall decrease in throughput. Given these confoundingly interrelated factors, along with latent variables such as the acceptance rate (which is unknown before verification and depends on the input prompts), we adopt a bandit-based approach to model the current throughput as the reward. Specifically, we employ UCBSPEC to dynamically adjust the hyperparameter $\gamma$, the speculation length, to maximize the throughput, i.e., the number of generated tokens per second. We set the maximum speculation length $L$ as 4, and $\gamma$ takes values in $\{0, \ldots, 4\}$ where $\gamma = 0$ corresponds to the canonical decoding (Algorithm 1). As the first experiment suggests UCBSPEC is more in line with the real-life speculative decoding environment than EXP3SPEC, we only evaluate UCBSPEC in this experiment. The experiments are conducted on a single A100.

Specifically, we use LLaMA3-8B-Instruct and Qwen2-7B-Instruct as the target models and adopt Eagle-1 (Li et al., 2024a), the current state-of-the-art model, as the draft model. We do not use Eagle-2 (Li et al., 2024b) because it does not support batch inference. For evaluation, we adopt Alpaca (Taori et al., 2023) as the test set, as it covers various topics, thereby simulating a realistic setting with diverse acceptance rates. To approximate real-world conditions, we randomly sample prompts from the test set to form a batch for inference, with batch sizes ranging from 1 to 50. As

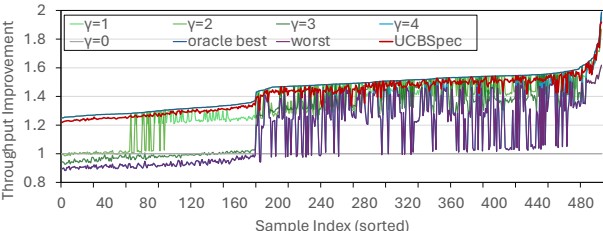

(a) Target model: LLaMA3

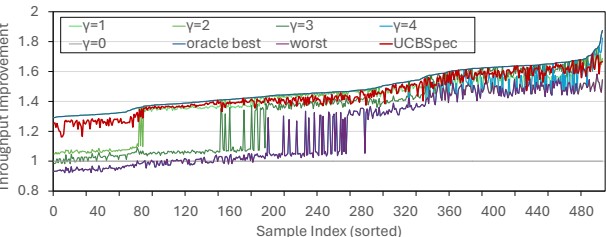

(b) Target model: Qwen2

*Figure 3.* We compare throughtput improvements with different speculative decoding lengths $\gamma \in [4]$ and the canonical decoding ($\gamma = 0$). The performance of UCBSPEC approaches that of the best hyperparameter across all samples for both target models LLaMA3 and Qwen2. The sample indices are sorted according to the best arm improvement for a clear demonstration.

our evaluation metric, we measure the throughput improvement relative to the canonical decoding (non-speculative) baseline. Our result is averaged over 16 independent runs to smoothen the hardware-dependent factors.

**Experimental Results** The results are presented in Figure 3, where we reorder the 500 *sample indices* in ascending order of the performance of the best hyperparameter (blue line) for easy comparison. Otherwise, the lines in this figure will not be largely monotonic. Here the "worst" and "best" lines are calculated among results of $\gamma \in \{1, \cdots, 4\}$ in *hindsight*. Thus, we call the "best" line as the oracle best. **Firstly**, since the optimal hyperparameter $\gamma$ varies with different input prompts for either target model, fixing a single hyperparameter is suboptimal, e.g., in Figure 3 (b), the best hyperparameter changes from $\gamma = 1$ (light green) to $\gamma = 2$ (green) at sample index around 80; and the original Eagle-1 (Li et al., 2024a) ($\gamma = 4$ in purple) is even inferior to the canonical decoding ($\gamma = 0$ in grey) for sample indices less than 80. This necessitates the use of adaptive hyperparameter selection. **Next**, UCBSPEC demonstrates competitive throughput performance, outperforming the second-best hyperparameter in most cases and closely approaching the (varying) oracle best across experiments. These benefits are obtained thanks to the adaptivity of BANDITSPEC.

## 7. Conclusions and Discussions

In this work, we propose a MAB framework together with two hyperparameter selection algorithms that adaptively

choose appropriate hyperparameters to accelerate LLM inference under realistic assumptions. Both theoretical guarantees and extensive experiments are provided to demonstrate that adaptive speculative decoding via bandit algorithms can boost the performance of existing methods in a training-free manner. For future work, we would like to point some directions, improving the performance of the current algorithms.

Therefore, another direction is to design hyperparameter selection algorithms that can achieve the (near) optimal balance between these two goals based on practical needs.

**Structured Bandits** Our current framework is based on the standard $K$-armed bandit model. However, broader classes of bandit problems with additional structures—such as linear bandits (Abbasi-yadkori et al., 2011) and Lipschitz bandits (Magureanu et al., 2014)—can also be considered. This aligns more closely with practical scenarios, where the number of hyperparameters can be large, and the value of $K$ may be very high when modeling the problem as a $K$-armed MAB. By leveraging such structures in MABs, we can expect to identify better hyperparameters more efficiently, thereby further accelerating the optimization process.

**Robust bandits and Non-stationary bandits** As indicated by the experimental result, the real-life speculative decoding environment is closer to the stochastic reward case (Assumption 4.1) than the adversarial reward case (Assumption 5.1). Therefore, one direction for future work is to consider the settings "in between", e.g., robust bandits in the presence of adversarial corruptions (Ding et al., 2022; Zhong et al., 2021), or non-stationary bandits (Cao et al., 2019; Besbes et al., 2014; Hou et al., 2024) where the mean number of accepted tokens can vary across time. These settings are more benign than the adversarial reward assumption and can be exploited to accelerate the inference.

**Contextual Bandits** Another direction is to explore contextual bandits, where the environment reveals additional information that can be leveraged to reduce the learning burden (Luo et al., 2018; Kato & Ariu, 2021).

## Impact Statement

This paper presents work whose goal is to advance the field of Machine Learning. There are many potential societal consequences of our work, none which we feel must be specifically highlighted here.

**Acknowledgements:** This work is supported by funding from the Singapore Ministry of Education Academic Research Fund (AcRF) Tier 1 grants under grant numbers A-8002934-00-00 and A-8000980-00-00. This research is also supported by the National Research Foundation, Singapore under its AI Singapore Programme (AISG Award No: AISG2-PhD-2023-08-044T-J), and is part of the programme DesCartes which is supported by the National Research

Foundation, Prime Minister's Office, Singapore under its Campus for Research Excellence and Technological Enterprise (CREATE) programme.

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

# A. Related Works

**Speculative Decoding** Speculative decoding is proposed in Leviathan et al. (2023); Chen et al. (2023), where the draft model only generates a single chain of draft tokens. Then a line of works extends the chain structure to the tree structure (Miao et al., 2023; Cai et al., 2024; Du et al., 2024; Li et al., 2024a; Hu & Huang, 2024). In these works, the draft tokens are organized as a connected tree. To further improve the number of accepted tokens, previous works propose to generate tokens in a batch manner, i.e., the draft tokens are organized as multiple disconnected parts. SpecTr (Sun et al., 2024c) views this problem from the optimal transport perspective and derives the algorithm that is optimal up to a multiplicative constant. Khisti et al. (2025) derives the canonical form of this problem and design the relaxed optimization algorithms. All these algorithms verify the draft tokens in a token by token manner. Sun et al. (2024b) proposes to verify all the draft tokens as a whole block, which further boosts the acceleration ratio. Sun et al. (2024a) proposes to fit the speculative deciding into a hierarchical structure, which multiple draft models with various sizes are generating and verifying tokens. The smaller model will generate more tokens. This fine-grained behavior improve the overall performance of the system. Liu et al. (2023) design algorithms to update the draft model parameters in an online manner, which makes the draft model adaptive to the current context. Liu et al. (2024) and Huang et al. (2024) aim to optimize the speculative length in a training and training-free manner (more discussions on SpecDec++ (Huang et al., 2024) are provided in Appendix B.4). Chen et al. (2024) optimizes the hyperparameters related to the hardware by dynamic programming in an offline manner. We also note that there are a series of non-parametric speculative decoding algorithms (Hu et al., 2024; Oliaro et al., 2024), i.e., the draft model itself does not require any training procedures. Yin et al. (2024) derives the theoretical analysis of the speculative decoding.

**Multi-Armed Bandit** The multi-armed bandit problem is a fundamental topic in decision theory and reinforcement learning, with various algorithms developed to address the exploration-exploitation trade-off. The standard stochastic $K$-armed bandit problem is firstly introduced by Robbins (1952) and then studied by Lai & Robbins (1985). There has been a major theoretical advancement with the introduction of Upper Confidence Bound (UCB) algorithms (Auer et al., 2002b). Various algorithms have been proposed to achieve improved theoretical guarantees and practical performance since then (Garivier & Cappé, 2011; Bubeck et al., 2012). Beyond UCB-type algorithms, sampling-based algorithms, such as Thompson Sampling (Agrawal & Goyal, 2012; 2017; Russo et al., 2017) and sampling via bootstrap (Kveton et al., 2019; Wan et al., 2023), have also exhibited strong empirical performance with provable regret bounds. Furthermore, the problem has been extended to the adversarial settings where the rewards are no longer stochastic (Auer et al., 2002a; Bubeck et al., 2012). We refer to Lattimore & Szepesvári (2020) for a comprehensive introduction of the Multi-Armed Bandit problem.

# B. Additional Discussions and Remarks

## B.1. Discussion on the Assumptions

On the theoretical side, the stationary mean assumption (Assumption 4.1) is strictly weaker than the i.i.d. assumption. In particular, the number of accepted tokens can depend on the generated tokens. Therefore, this assumption is aligned with real-world scenarios in which different decoding steps are correlated. Furthermore, the basic Multi-Armed Bandits (MAB) model can be generalized to contextual bandits and non-stationary bandits. The proposed BANDITSPEC framework provides a basic template to apply these more general MAB setups to speculative decoding. Our formulations under the stationary/adversarial mean assumptions are just basic setups and we leave the more general/elaborate setups as future research.

On the experimental side, our experimental results (Table 1) indicate, the performance of UCBSPEC significantly outperforms one of the best speculative decoding methods, Eagle-2 (Li et al., 2024). This corroborates the stationary mean assumption in our formulation.

## B.2. Discussion on UCBSPEC

We comment that UCBSPEC is among the simplest UCB-type algorithms in the sense that only the empirical means and UCB's need to be maintained, and the hyperparameter to be selected can be directly determined via the UCB's.

In contrast, Thompson Sampling (Agrawal & Goyal, 2012) and KL-UCB (Garivier & Cappé, 2011) generally achieve better empirical regret bounds than UCB1 (Auer et al., 2002b). However, Thompson Sampling requires maintaining the posterior distribution and sampling from it to select the arm to pull; whereas KL-UCB involves solving an optimization problem for arm selection. These steps add additional complexity to the algorithms.

Given our goal of accelerating LLM inference, the simplicity of UCB1 is more in line with this objective. Therefore, we propose UCBSPEC, which redesigns the confidence radius and the stopping rule of UCB1 to adapt specifically to the speculative decoding application.

### B.3. Discussion on Adaptive Adversary

We consider the oblivious adversary in this paper where the numbers of accepted tokens generated by all hyperparameters at all time steps, i.e., $\{y_{i,t}\}_{i\in[K],t\in\mathbb{N}}$, are fixed before the decoding process starts. One may be interested in considering the *adapative* adversary, where the environment (adversary) can choose the number of accepted tokens generated by $S_{I_t}$ based on (part of) the history information $\mathcal{H}_{t-1}$ and $I_t$ (Arora et al., 2012). This adversary is more malicious than the oblivious one and the regret is expected to be even larger than the current one in Theorem 5.3. As our empirical experiments suggest that the practical scenario aligns more closely with the stochastic reward assumption (Assumption 4.1) and deviates from the oblivious adversarial reward assumption (Assumption 5.1), we believe it is unnecessary to consider the adaptive adversarial reward.

### B.4. Discussion on SpecDec++ (Huang et al., 2024)

We compare the proposed methods with an existing adaptive speculative decoding method, SpecDec++ (Huang et al., 2024), in this section.

SpecDec++ (Huang et al., 2024) is adaptive in choosing the speculation length, achieving good performance compared to the vanilla speculative decoding method (Leviathan et al., 2023; Chen et al., 2023). It trains an acceptance probability prediction head and stops drafting new tokens when the predicted rejection probability reaches certain threshold.

We compare it with the proposed methods as follows:

- Firstly, we highlight that our proposed method is *training-free* which can be deployed easily along with *existing off-the-shelf methods*. In contrast, SpecDec++ focuses on *training* of an acceptance prediction head. Currently, SpecDec++ is only available when using LLaMA-2-Chat-7B as the draft model and LLaMA-2-Chat-70B as the target model (bfloat 16).

- Secondly, the proposed BANDITSPEC framework considers the more general hyperparameter selection problem that goes beyond merely the speculation length. Therefore, it is "orthogonal" to SpecDec++ in the sense that any methods with (or without) SpecDec++ can also be candidates for the hyperparameter in our framework, e.g., {Eagle-2, LLaMA-2-Chat-7B} with SpecDec++ can also be regarded as arms (if they are available).

## C. Additional Details

In this section, we provide more details that complement the main paper.

### C.1. Vanilla Speculative Decoding

For completeness, we present and describe the vanilla speculative decoding algorithm (Leviathan et al., 2023; Chen et al., 2023) in Algorithm 6 in this section.

We introduce some notations first. For any nonnegative function $f : \mathcal{X} \to \mathbb{R}_+$ with $\sum_{x\in\mathcal{X}} f(x) > 0$, we define the distribution induced by it as $\text{Norm}(f(\cdot)) = f(\cdot)/\sum_{x'\in\mathcal{X}} f(x')$. The positive part of a function $f$ is denoted as $[f(\cdot)]_+ = \max\{0, f(\cdot)\}$.

Speculative decoding implements a draft model $Q$ to generate draft tokens and let the target model $P$ verify them in *parallel*. In practice, the draft model is much smaller than the target model. Thus, the draft token generation (Line 1) can be achieved in a short time. Then we let the target model only forward inference *once* with these draft tokens as inputs (Line 2). The verification procedures (Lines 4 to 9) are designed to guarantee that the output tokens $x_{1:\tau+1}$ is distributed as the target model $P$. Here, the additional $(\tau+1)$-st accepted token is also called the *bonus token*.

**Algorithm 6** Vanilla Speculative Decoding

**Inputs:** base model $P$, draft model $Q$, prefix pt, maximum speculation length $L$

**Procedures:**

1: Generate $L$ draft tokens $\tilde{x}_{1:L}$ via $\tilde{x}_i \sim Q(\cdot \,|\, \mathrm{pt}, \tilde{x}_{1:i-1})$ for $i \in [L]$.
2: Set $\tau = 0$, and calculate the values of $P(\tilde{x}_i \,|\, \mathrm{pt}, \tilde{x}_{1:i-1})$ for $i \in [L]$ in parallel.
3: **for** $k = 1, \ldots, L$ **do**
4:     Sample $r_i \sim \mathrm{Unif}([0,1])$.
5:     **if** $r_i \leq \min\{1, P(\tilde{x}_i \,|\, \mathrm{pt}, \tilde{x}_{1:i-1})/Q(\tilde{x}_i \,|\, \mathrm{pt}, \tilde{x}_{1:i-1})\}$ **then**
6:         Set $\tau = i$ and $x_i = \tilde{x}_i$.
7:     **else**
8:         Sample $x_i \sim \mathrm{Norm}\big([P(\cdot \,|\, \mathrm{pt}, \tilde{x}_{1:i-1}) - Q(\cdot \,|\, \mathrm{pt}, \tilde{x}_{1:i-1})]_+\big)$.
9:         **break.**
10:     **end if**
11: **end for**
12: **if** $\tau = L$ **then** sample $x_{L+1} \sim P(\cdot \,|\, \mathrm{pt}, x_{1:L})$.
13: **return** $x_{1:\tau+1}$.

## C.2. Dynamics of MAB

We provide a description of the dynamics of MAB in Algorithm 7.

**Algorithm 7** Dynamics of MAB

**Inputs:** $K$ **arms, time horizon** $T$.

1: $\mathcal{H}_0 = \emptyset$.
2: **for** $t = 1, 2, \ldots, T$ **do**
3:     Agent adopts an algorithm to select $I_t$ based on $\mathcal{H}_{t-1}$.
4:     Environment reveals the reward $X_{I_t,t}$ to the agent.
5:     $\mathcal{H}_t = \mathrm{concat}(\mathcal{H}_{t-1}, (I_t, X_{I_t,t}))$.
6: **end for**

The goal is to minimize the cumulative regret

$$\max_{i \in [K]} \mathbb{E}\left[\sum_{t=1}^{T} X_{i,t}\right] - \mathbb{E}\left[\sum_{t=1}^{T} X_{I_t,t}\right],$$

where the expectation is taken w.r.t. the randomness in the rewards (for the stochastic setup) and the possible internal randomness in the arm selection algorithm.

## C.3. Full Description of BANDITSPEC with UCBSPEC

We provide the full description of BANDITSPEC(ALG) with ALG = UCBSPEC in Algorithm 8.

---

**Algorithm 8** BANDITSPEC(UCBSPEC) (Full version of UCBSPEC)

---

**Inputs:** initial prompt $\text{pt}_0 = \text{pt} \in \mathcal{X}^*$, bandit configuration $\nu = (P, \mathcal{S} = \{S_i\}_{i \in [K]}, L)$.
**Procedures:**

1: $t = 0, \mathcal{H}_0 = \emptyset, I_0 = 1, x_{I_0,0} = \emptyset$.
2: **while** EOS $\notin x_{I_t,t}$ **do**
3:     $t = t + 1$
4:     **if** $t \le K$ **then**
5:         $I_t = t$. (Round-Robin)
6:     **else**
7:         Select index $I_t = \text{argmax}_{i \in [K]} UCB_{i,t-1}$.
8:     **end if**
9:     Observe $X_{I_t,t} = x_{I_t,t} = \text{SPECDECSUB}(\text{pt}_{t-1}, P, S_{I_t}, L)$   and   $Y_{I_t,t} = y_{I_t,t} = \text{len}(X_{I_t,t})$.
10:    $\text{pt}_t = \text{concat}(\text{pt}_{t-1}, X_{I_t,t}), \mathcal{H}_t = \text{concat}(\mathcal{H}_{t-1}, (I_t, X_{I_t,t}))$.
11:    Update the statistics $\{\hat{\mu}_{i,t}\}_{i \in [K]}, \{\text{cr}_{i,t}\}_{i \in [K]}$, where

$$n_{i,t} = \sum_{s=1}^{t} \mathbb{1}\{I_s = i\}, \ \hat{\mu}_{i,t} = \frac{\sum_{s=1}^{t} Y_{i,s} \mathbb{1}\{I_s = i\}}{n_{i,t}},$$

$$\text{cr}_{i,t} = \frac{L}{2} \sqrt{\frac{1 + n_{i,t}}{n_{i,t}^2} \left(1 + 2\log \frac{Kt^2(1 + n_{i,t})^{\frac{1}{2}}}{\delta}\right)},$$

$$\text{UCB}_{i,t} = \hat{\mu}_{i,t} + \text{cr}_{i,t}.$$

12: **end while**
13: **return** $t, \text{pt}_t$

---

## C.4. Full Description of BANDITSPEC with EXP3SPEC

We provide the full description of BANDITSPEC(ALG) with ALG = EXP3SPEC in Algorithm 9.

---

**Algorithm 9** BANDITSPEC(EXP3SPEC) (Full version of EXP3SPEC)

---

**Inputs:** initial prompt $\text{pt}_0 = \text{pt} \in \mathcal{X}^*$, bandit configuration $\nu = (P, \mathcal{S} = \{S_i\}_{i \in [K]}, L)$, learning rates $\eta_t = \sqrt{\frac{\log K}{t \cdot K}}, t \in \mathbb{N}$.
**Procedures:**

1: $t = 0, \mathcal{H}_0 = \emptyset, I_0 = 1, x_{I_0,0} = \emptyset$.
2: **while** EOS $\notin x_{I_t,t}$ **do**
3:     $t = t + 1$
4:     Set probability vector $p_t \in \mathbf{\Delta}_{[K]}$ with

$$p_{t,i} = \frac{\exp\left(-\eta_t \sum_{s=1}^{t-1} \widehat{Z}_{i,s}\right)}{\sum_{j=1}^{K} \exp\left(-\eta_t \sum_{s=1}^{t-1} \widehat{Z}_{j,s}\right)} \quad, \forall i \in [K].$$

5:     Select a hyperparameter index $I_t \sim p_t$.
6:     Observe $X_{I_t,t} = x_{I_t,t} = \text{SPECDECSUB}(\text{pt}_{t-1}, P, S_{I_t}, L)$ and $y_{I_t,t} = \text{len}(X_{I_t,t})$.
7:     $\text{pt}_t = \text{concat}(\text{pt}_{t-1}, X_{I_t,t}), \mathcal{H}_t = \text{concat}(\mathcal{H}_{t-1}, (I_t, X_{I_t,t}))$.
8:     Set the statistics

$$\widehat{Z}_{i,t} = \mathbb{1}\{i = I_t\} \cdot \frac{L + 1 - y_{i,t}}{L \cdot p_{t,i}}, \quad \forall i \in [K].$$

9: **end while**
10: **return** $t, \text{pt}_t$

---

# D. Proofs of Main Results

## D.1. Proof of Proposition 3.1

To prove Proposition 3.1, we note that we only need to prove

$$\mathrm{pt}_{\mathrm{ST(ALG)}} \overset{d}{=} \mathrm{pt}_{\tau_c}, \quad \text{and} \quad \frac{\mathrm{len}(\mathrm{pt}_{\mathrm{ST(ALG)}})}{L+1} \leq \mathrm{ST(ALG)} \leq \mathrm{len}(\mathrm{pt}_{\mathrm{ST(ALG)}}), \quad a.s.$$

The other results are implied by these two results. For equality, we note that this is already proved by Theorem 1 in Yin et al. (2024), where the equality holds for any specification of the hyperparameters. For the inequality, we note that each implementation of SPECDECSUB generates at least one token and at most $L+1$ tokens. Thus, the inequality holds almost surely.

## D.2. Proof of Theorem 4.3

**Theorem 4.3** (Upper Bound). *Under Assumptions 4.1 and 4.2, given any prompt* $\mathrm{pt} \in \mathcal{X}^*$ *and bandit configuration* $\nu = (P, \mathcal{S} = \{S_i\}_{i\in[K]}, L)$, *the expected stopping time regret of Algorithm 3 with* $\mathrm{ALG} = $ *Algorithm 4 (*UCBSPEC*) is upper bounded as*

$$\mathrm{Reg}(\mathrm{ALG}, \mathrm{pt}, \nu) = O\Big(\mathrm{H}(\mathrm{pt}, \nu) \cdot L^2 \log \mathbb{E}[\mathrm{len}(\mathrm{pt}_{\tau_c})]\Big).$$

*Proof of Theorem 4.3.* Our proof of Theorem 4.3 consists of three steps.

- Reward and Stopping time decomposition.

- Construction of the high probability event.

- Concluding the proof.

As we mentioned in Section 4, the main difference lies at the two aspects:
**Firstly**, the stopping time is now a random, which depends on the generated tokens. This cause trouble when we decompose the reward/regret, as both the rewards and time horizon depend on the history. We tackle this problem in Step 1 by making use of the martingale structure of the rewards sequence.
**Secondly**, we consider the problem under Assumption 4.1, where the number of accepted tokens can be dependent. This is practical as LLM generates tokens in an autoregressive manner. In contrast, under the commonly seen assumption for the $K$-armed MAB, the rewards are i.i.d. and can be regarded as they have been sampled before the algorithm starts (see Chapter 4 in Lattimore & Szepesvári (2020)). Thus, Chernoff-Hoeffding bound (Lemma F.2) can be directly applied, which cannot be used under Assumption 4.1. We solve this problem in Step 2, by adopting the so-called self-normalized confidence bounds (Abbasi-yadkori et al., 2011).

**Step 1: Reward and Stopping time decomposition.**

By the property of speculative decoding in (2) and (11), for any algorithm ALG, [3]

$$\mathbb{E}\big[\mathrm{len}(\mathrm{pt}_{\tau_c})\big] = \mathbb{E}[\mathrm{len}(\mathrm{pt}_{\mathrm{ST(ALG)}})] = \mathbb{E}\Bigg[\sum_{t=1}^{\mathrm{ST(ALG)}} Y_{I_t,t}\Bigg]$$

We wish to decompose the expected total token sequence in terms of each hyperparameter $i \in [K]$ in the first step, i.e.,

$$\mathbb{E}\Bigg[\sum_{t=1}^{\mathrm{ST(ALG)}} Y_{I_t,t}\Bigg] = \sum_{i=1}^{K} \mu_i \cdot \mathbb{E}\big[n_{i,\mathrm{ST(ALG)}}\big].$$

---

[3]We clarify that only the tokens up to the EOS token will be appended to the prefix token sequence in practice. Therefore, the actual number of accepted tokens is between $[1, Y_{I_{\mathrm{ST(ALG)}},\mathrm{ST(ALG)}}]$. While this mismatch can be solved, using $Y_{I_{\mathrm{ST(ALG)}},\mathrm{ST(ALG)}}$ as the number of accepted tokens in the last round will introduce an error of at most $L$ to the token sequence length $\mathrm{len}(\mathrm{pt}_{\mathrm{ST(ALG)}})$, which is an error of at most 1 to the stopping time $\mathrm{ST(ALG)}$. This error is negligible compared to the other values in the stopping time regret. Therefore, we assume the EOS token only appears at the end of accepted tokens for the sake of simplicity.

The standard regret analysis adopts Wald's equation to decompose the expected cumulative regret, or equivalently the stopping time. However, as both the stopping time $\text{ST}(\text{ALG})$ and $Y_{I_t,t}$ depends on the history under our problem setup, Wald's equation fails. We propose a new and general approach to decompose the reward.

• Step 1.1: We first prove that $M_n := \sum_{t=1}^{n} Y_{I_t,t} - \mu_{I_t}, n = 0, 1, 2, \ldots$ is a martingale with respect to $\{\mathcal{F}_n\}_{n=0}^{\infty}$, where $M_0 := 0, \mathcal{F}_n := \sigma(\mathcal{H}_n)$.

By the definition of martingale, we only need to show (1) $\mathbb{E}[|M_n|] < \infty$, and (2) $\mathbb{E}[M_{n+1}|\mathcal{F}_n] = M_n$.

(1) $\underline{\mathbb{E}[|M_n|] < \infty}$: As the number of the accepted tokens at each round is bounded as $Y_{I_t,t} \in [1, L+1]$ almost surely and $\mu_{I_t} \in [1, L+1]$, we have $|Y_{I_t,t} - \mu_{I_t}| \le L$. Then the triangular inequality $|M_n| \le \sum_{t=1}^{n} |Y_{I_t,t} - \mu_{I_t}| \le L \cdot n < \infty$ indicates that

$$\mathbb{E}[|M_n|] < \infty. \tag{7}$$

(2) $\underline{\mathbb{E}[M_{n+1}|\mathcal{F}_n] = M_n}$: The conditional expectation of $M_{n+1}$ can be calculated via tower property as

$$\mathbb{E}[M_{n+1} \mid \mathcal{F}_n] = M_n + \mathbb{E}\left[\mathbb{E}\left[Y_{I_{n+1},n+1} - \mu_{I_{n+1}} \mid \mathcal{H}_n, I_{n+1}\right] \mid \mathcal{H}_n\right] = M_n, \tag{8}$$

where the last equality results from Assumption 4.1.

Based on (7) and (8), $M_n, n = 0, 1, 2, \ldots$ is a martingale with respect to $\{\mathcal{F}_n\}_{n=0}^{\infty}$.

• Step 1.2: We then prove $\mathbb{E}[M_{\text{ST}(\text{ALG})}] = 0$ via Doob's optional stopping lemma (Lemma F.1).

We have already showed that $\sum_{t=1}^{n} Y_{I_t,t} - \mu_{I_t}, n = 1, 2, \ldots$ is a martingale with respect to $\{\mathcal{H}_n\}_{n=0}^{\infty}$. In order to apply Lemma F.1, we firstly verify the prerequisites listed in Lemma F.1 condition (b): (1) $\mathbb{E}[\text{ST}(\text{ALG})] < \infty$, and (2) there exists $c \in \mathbb{R}$, such that $\mathbb{E}[|M_t - M_{t-1}| \mid \mathcal{F}_{t-1}] \le c$ almost surely for $t \le \text{ST}(\text{ALG})$.

(1) $\underline{\mathbb{E}[\text{ST}(\text{ALG})] < \infty}$: According to the property of speculative decoding (3) and Assumption 4.2, we have that $\mathbb{E}[\text{ST}(\text{ALG})] \le \mathbb{E}[\text{len}(\text{pt}_{\tau_c})] < \infty$.

(2) $\underline{\mathbb{E}[|M_t - M_{t-1}| \mid \mathcal{F}_{t-1}] \le c}$: As $M_t - M_{t-1} = Y_{I_t,t} - \mu_{I_t}$ and $|Y_{I_t,t} - \mu_{I_t}| \le L$ almost surely, it holds that $\mathbb{E}[|M_t - M_{t-1}| \mid \mathcal{F}_{t-1}] \le L$. Taking $c = L$ finishes the verification.

Therefore, condition (b) in Lemma F.1 is satisfied and we obtain

$$\mathbb{E}\left[\sum_{t=1}^{\text{ST}(\text{ALG})} Y_{I_t,t} - \mu_{I_t}\right] = 0. \tag{9}$$

Step 1.3: We show that $\mathbb{E}\left[\sum_{t=1}^{\text{ST}(\text{ALG})} Y_{I_t,t}\right] = \sum_{i=1}^{K} \mu_i \cdot \mathbb{E}\left[n_{i,\text{ST}(\text{ALG})}\right]$.

We firstly note that

$$\mathbb{E}\left[\left|\sum_{t=1}^{\text{ST}(\text{ALG})} Y_{I_t,t}\right|\right] \le (L+1)\mathbb{E}[\text{ST}(\text{ALG})] < \infty \quad \text{and} \quad \mathbb{E}\left[\left|\sum_{t=1}^{\text{ST}(\text{ALG})} \mu_{I_t}\right|\right] \le (L+1)\mathbb{E}[\text{ST}(\text{ALG})] < \infty,$$

so the expectations of $\sum_{t=1}^{\text{ST}(\text{ALG})} Y_{I_t,t}$ and $\sum_{t=1}^{\text{ST}(\text{ALG})} \mu_{I_t}$ exist and are finite (integrable).

Furthermore, we have

$$\mathbb{E}\left[\sum_{t=1}^{\text{ST}(\text{ALG})} \mu_{I_t}\right] = \mathbb{E}\left[\sum_{i=1}^{K}\sum_{t=1}^{\text{ST}(\text{ALG})} \mathbb{1}\{I_t = i\} \cdot \mu_i\right] = \sum_{i=1}^{K} \mu_i \cdot \mathbb{E}\left[n_{i,\text{ST}(\text{ALG})}\right], \tag{10}$$

where $n_{i,\text{ST}(\text{ALG})} = \sum_{t=1}^{\text{ST}(\text{ALG})} \mathbb{1}\{I_t = i\}$ by definition. Because $\sum_{t=1}^{\text{ST}(\text{ALG})} Y_{I_t,t}$ and $\sum_{t=1}^{\text{ST}(\text{ALG})} \mu_{I_t}$ are integrable, (9) and (10) imply

$$\mathbb{E}\left[\sum_{t=1}^{\text{ST}(\text{ALG})} Y_{I_t,t}\right] = \mathbb{E}\left[\sum_{t=1}^{\text{ST}(\text{ALG})} Y_{I_t,t} - \mu_{I_t}\right] + \mathbb{E}\left[\sum_{t=1}^{\text{ST}(\text{ALG})} \mu_{I_t}\right] = \sum_{i=1}^{K} \mu_i \cdot \mathbb{E}\left[n_{i,\text{ST}(\text{ALG})}\right]. \tag{11}$$

This equality decomposes the cumulative reward (and stopping time) in terms of the arms.

**Step 2: Construction of the high probability event.**

We then derive the concentration property for the number of accepted tokens. Define the good events:

$$\mathcal{E}_t := \big\{ \hat{\mu}_{i,t} \in [\mu_i - \mathrm{cr}_{i,t}, \mu_i + \mathrm{cr}_{i,t}], \forall i \in [K] \text{ at round } t \big\}.$$

Since random variables supported on $[a, b]$ is $(b - a)^2/4$-sub-Gaussian and $1 - \mu_{I_t} \le Y_{I_t, t} - \mu_{I_t} \le L + 1 - \mu_{I_t}$ under our problem setup. According to Lemma F.3, we obtain

$$\mathbb{P}(\mathcal{E}_t) \ge 1 - \frac{\delta}{t^2} \quad \text{and} \quad \sum_{t=K+1}^{\infty} \mathbb{P}(\mathcal{E}_t^c) \le \frac{\pi^2 \delta}{6}, \tag{12}$$

where $\delta$ is a confidence parameter that will be specified later. We remark that Lemma F.3 from Abbasi-yadkori et al. (2011) adopts a self-normalized concentration bound for the martingale sequence, which generalizes the standard i.i.d. reward assumption in the $K$-armed MAB problem.

As a result, we can now bound the number of times arm $i$ is pulled at any round $t \ge K$. Conditional on the good event $\mathcal{E}_t$, we have $\hat{\mu}_{i,t} \in [\mu_i - \mathrm{cr}_{i,t}, \mu_i + \mathrm{cr}_{i,t}]$ and arm $i$ will not be pulled if $\mathrm{cr}_{i,t} < \Delta_i/2$. By adopting Lemma F.4, when arm $i$ is selected at time $t + 1$, it must hold that

$$n_{i,t} \le 4 + \frac{2L^2}{\Delta_i^2} \Big( 1 + 2 \log \frac{LK \cdot t^2}{\Delta_i \delta} \Big). \tag{13}$$

**Step 3: Concluding the proof.**

According to Step 1,

$$\mathbb{E}\big[\mathrm{len}(\mathrm{pt}_{\tau_c})\big] = \mathbb{E}[\mathrm{len}(\mathrm{pt}_{\mathrm{ST}(\mathtt{ALG})})] = \mathbb{E}\Big[ \sum_{t=1}^{\mathrm{ST}(\mathtt{ALG})} Y_{I_t, t} \Big] = \sum_{i=1}^{K} \mu_i \cdot \mathbb{E}\big[ n_{i,\mathrm{ST}(\mathtt{ALG})} \big].$$

Therefore,

$$\begin{aligned}
\mathrm{Reg}(\mathtt{ALG}) &= \frac{1}{\mu_{i^*}} \bigg( \sum_{i \in [K]} \mu_i \cdot \mathbb{E}\big[ n_{i,\mathrm{ST}(\mathtt{ALG})} \big] + \sum_{i \in [K]} \Delta_i \cdot \mathbb{E}\big[ n_{i,\mathrm{ST}(\mathtt{ALG})} \big] - \mu_{i^*} \cdot \mathbb{E}\left[ \mathrm{ST}(\mathtt{ALG}_{i^*}) \right] \bigg) \\
&= \sum_{i \neq i^*} \frac{\Delta_i}{\mu_{i^*}} \cdot \mathbb{E}[n_{i,\mathrm{ST}(\mathtt{ALG})}]. \tag{14}
\end{aligned}$$

Under the UCBSPEC algorithm, we have

$$\begin{aligned}
&\sum_{i \neq i^*} \frac{\Delta_i}{\mu_{i^*}} \cdot \mathbb{E}\big[ n_{i,\mathrm{ST}(\mathtt{ALG})} \big] \\
&= \sum_{i \neq i^*} \frac{\Delta_i}{\mu_{i^*}} \cdot \mathbb{E}\Big[ \sum_{t=K+1}^{\mathrm{ST}(\mathtt{ALG})} \mathbb{1}\Big\{ I_t = i, n_{i,t-1} \le 4 + \frac{2L^2}{\Delta_i^2} \Big( 1 + 2 \log \frac{LKt^2}{\Delta_i \delta} \Big) \Big\} \Big] \\
&\quad + \sum_{i \neq i^*} \frac{\Delta_i}{\mu_{i^*}} \cdot \mathbb{E}\Big[ \sum_{t=K+1}^{\mathrm{ST}(\mathtt{ALG})} \mathbb{1}\Big\{ I_t = i, n_{i,t-1} > 4 + \frac{2L^2}{\Delta_i^2} \Big( 1 + 2 \log \frac{LK(t-1)^2}{\Delta_i \delta} \Big) \Big\} \Big] + K \\
&\le \sum_{i \neq i^*} \frac{\Delta_i}{\mu_{i^*}} \cdot \mathbb{E}\Big[ 4 + \frac{2L^2}{\Delta_i^2} \Big( 1 + 2 \log \frac{LK \cdot (\mathrm{ST}(\mathtt{ALG}))^2}{\Delta_i \delta} \Big) \Big] + \mathbb{E}\Big[ \sum_{t=K+1}^{\mathrm{ST}(\mathtt{ALG})} \mathbb{1}\{ \mathcal{E}_t^c \} \Big] + K \\
&\le \sum_{i \neq i^*} \frac{\Delta_i}{\mu_{i^*}} \cdot \Big( 4 + \frac{2L^2}{\Delta_i^2} \Big( 1 + 2 \log \frac{LK \cdot (\mathbb{E}[\mathrm{ST}(\mathtt{ALG})])^2}{\Delta_i \delta} \Big) \Big) + \sum_{t=K+1}^{\infty} \mathbb{P}(\mathcal{E}_t^c) + K, \\
&\le \sum_{i \neq i^*} \frac{\Delta_i}{\mu_{i^*}} \cdot \Big( 4 + \frac{2L^2}{\Delta_i^2} \Big( 1 + 2 \log \frac{LK \cdot (\mathbb{E}[\mathrm{len}(\mathrm{pt}_{\tau_c})])^2}{\Delta_i \delta} \Big) \Big) + \frac{\pi^2 \delta}{6} + K,
\end{aligned}$$

where the first inequality results from (13), the second inequality utilizes Jensen's inequality, and the last inequality adopts the property of speculative decoding (3) and the upper bound on the error probability of good event (12) in Step 2. Taking $\delta = 1/2$ in the above bound concludes the proof of this theorem. □

### D.3. Proof of Theorem 5.3

Fix any $b \in [K]$, the baseline algorithm is set to be $\mathtt{ALG}_b$, i.e., Algorithm 3 implements Line 4 of Algorithm 3 with hyperparameter $S_b$ only. Let $\mathtt{ALG} = \mathtt{EXP3SPEC}$. we assume the $\mathtt{BANDITSPEC(ALG)}$ repeats the while loop in Algorithm 3 until $\max\{\mathrm{ST(ALG)}, \mathrm{ST(ALG}_b)\}$. To avoid any confusion, we restate the algorithm for the purpose of analysis in Algorithm 10. Algorithm 10 takes $\mathtt{ALG}$ and $\mathtt{ALG}_b$ as an input and stops until $\mathrm{pt}_{\mathrm{ST(ALG)}}$ and $\mathrm{pt}_{\mathrm{ST(ALG}_b)}$ are generated. The two token sequences up to EOS token are output at the end of the algorithm.

---

**Algorithm 10** BANDITSPEC(EXP3SPEC) (For analysis purpose)

---

**Inputs:** initial prompt $\mathrm{pt}_0 = \mathrm{pt} \in \mathcal{X}^*$, speculative decoding configuration $\nu = (P, \mathcal{S} = \{S_i\}_{i \in [K]}, L)$, stopping time $\tau = \infty$, baseline hyperparameter $S_b$, initial prompt $\mathrm{pt}_0^b = \mathrm{pt}$, stopping time $\tau^b = \infty$.

**Procedures:**

1: $t = 0, \mathcal{H}_0 = \emptyset, I_0 = 1, x_{I_0,0} = \emptyset, x_{b,0}^b = \emptyset$.
2: **while** $\tau = \infty$ or $\tau^b = \infty$ **do**
3:     $t = t + 1$
4:     // Procedures of the original EXP3SPEC
5:     **if** $\tau = \infty$ and $\mathrm{EOS} \in x_{I_{t-1},t-1}$ **then**
6:         $\tau = t - 1$ and $\mathrm{pt}_\tau = \mathrm{pt}_{t-1}$.
7:     **end if**
8:     Set probability vector $p_t \in \mathbf{\Delta}_{[K]}$ with

$$p_{t,i} = \frac{\exp\left(-\eta_t \sum_{s=1}^{t-1} \widehat{Z}_{i,s}\right)}{\sum_{j=1}^{K} \exp\left(-\eta_t \sum_{s=1}^{t-1} \widehat{Z}_{j,s}\right)} \text{ with learning rate } \eta_t = \sqrt{\frac{\log K}{t \cdot K}}, \ \forall i \in [K]. \tag{15}$$

9:     Select a hyperparameter index $I_t \sim p_t$.
10:     Observe $X_{I_t,t} = x_{I_t,t} = \mathrm{SPECDECSUB}(\mathrm{pt}_{t-1}, P, S_{I_t}, L)$ and $y_{I_t,t} = \mathrm{len}(X_{I_t,t})$.
11:     $\mathrm{pt}_t = \mathrm{concat}(\mathrm{pt}_{t-1}, X_{I_t,t}), \ \mathcal{H}_t = \mathrm{concat}(\mathcal{H}_{t-1}, (I_t, X_{I_t,t}))$.
12:     Set the statistics

$$\widehat{Z}_{i,t} = \mathbb{1}\{i = I_t\} \cdot \frac{L + 1 - y_{i,t}}{L \cdot p_{t,i}}, \quad \forall i \in [K]. \tag{16}$$

13:     // Procedures of the baseline $\mathtt{ALG}_b$
14:     **if** $\tau^b = \infty$ and $\mathrm{EOS} \notin x_{I_{t-1},t-1}^b$ **then**
15:         $\tau^b = t - 1$ and $\mathrm{pt}_{\tau^b}^b = \mathrm{pt}_{t-1}^b$.
16:     **end if**
17:     Observe $X_{b,t}^b = x_{I_t,t}^b = \mathrm{SPECDECSUB}(\mathrm{pt}_{t-1}^b, P, S_b, L)$ and $y_{b,t} = \mathrm{len}(X_{b,t}^b)$.
18:     $\mathrm{pt}_t^b = \mathrm{concat}(\mathrm{pt}_{t-1}^b, X_{b,t}^b)$.
19: **end while**
20: **return** $\mathrm{ST(ALG)} = \tau, \mathrm{pt}_{\mathrm{ST(ALG)}} = \mathrm{pt}_\tau$ and $\mathrm{ST(ALG}_b) = \tau^b, \mathrm{pt}_{\mathrm{ST(ALG}^b)} = \mathrm{pt}_{\tau^b}^b$.

---

**Theorem 5.3.** *Under Assumptions 4.2, 5.1 and 5.2, given any prompt* $\mathrm{pt} \in \mathcal{X}^*$ *and bandit configuration* $\nu = (P, \mathcal{S} = \{S_i\}_{i \in [K]}, L)$*, the expected stopping time regret of Algorithm 3 with* $\mathtt{ALG} = $*Algorithm 5 (EXP3SPEC),*

$$\mathrm{Reg}(\mathtt{ALG}, \mathrm{pt}, \nu) \leq 2L \cdot \min\left\{\sqrt{\mathrm{len}(\mathrm{pt}_{\tau_c})K \log K},\right.$$

$$\left. 2LK \log K + \sqrt{\min_{i \in [K]} \mathrm{ST(ALG}_i)K \log K}\right\}.$$

---

**Algorithm 11** Dynamics of the OLO Problem with Full Information Feedback

---

1: $\mathcal{H}_0 = \emptyset$.
2: **for** $t = 1, 2, \ldots, T$ **do**
3:     Selects $p_t \in \mathbf{\Delta}_{[K]}$ based on $\mathcal{H}_{t-1}$.
4:     Observes the loss vector $\ell_t$ and suffers loss $p_t^\top \ell_t$.
5:     $\mathcal{H}_t = \text{concat}(\mathcal{H}_{t-1}, (p_t, \ell_t))$.
6: **end for**

---

*Proof of Theorem 5.3.* For ease of presentation, we present Algorithm 10, where the while loop in BANDITSPEC is repeated until $\max\{\text{ST}(\texttt{ALG}), \text{ST}(\texttt{ALG}_i)\}$.

Our analysis of Algorithm 10 is novel compared to the standard analysis of EXP3 algorithm (Auer et al., 2002a; Lattimore & Szepesvári, 2020). It requires more technical manipulations due to the fact that the termination times of the baseline algorithm $\texttt{ALG}_i$ and $\texttt{ALG}$ are *different* and *random*, and that our goal is to minimize the stopping time regret (5).

For theoretical analysis, we regard Algorithm 10 as an instantiation of the Follow-the-Regularized-Leader (FTRL) algorithm (Gordon, 1999; Lattimore & Szepesvári, 2020).

The proof is decomposed into 5 steps:

- Connection between FTRL and Algorithm 10: we firstly introduce FTRL and the problem where it is applicapable. The shared features and differences are highlighted.

- Transformation of the stopping time regret: the stopping time regret is related to the regret under FTRL framework. In this case, the minimized regret by FTRL can be translated to the stopping time regret.

- Regret decomposition: the FTRL regret is decomposed for easier processing.

- Upper bound each term in the decomposed regret: we upper bound each term in the decomposed regret. The main difficulty is to deal with the difference in the time scales $\text{ST}(\texttt{ALG})$ and $\text{ST}(\texttt{ALG}_i)$ and the randomness in $\text{ST}(\texttt{ALG})$. Specifically, because both the loss vectors $\{\widehat{Z}_t\}_{t\in\mathbb{N}}$ and the stopping time $\text{ST}(\texttt{ALG})$ are random, taking expectation of the cumulative loss within $[\text{ST}(\texttt{ALG}_i), \text{ST}(\texttt{ALG})]$ is non-trivial. We devise Lemma D.1 and Lemma D.2 to deal with this problem.

- Conclusion of the stopping time regret: we aggregate the results in the previous steps and derive the final bound for the stopping time regret.

**Step 1: Connection between FTRL and Algorithm 5.**

We denote $\mathbf{\Delta}_{[K]}$ as the $K$-dimensional probability simplex.

FTRL is often used in the Online Learning Optimization Problem (OLO). We firstly provide a brief introduction to OLO that operates on $\mathbf{\Delta}_{[K]}$. Let $\ell_1, \ell_2, \ldots \in \mathbb{R}^K$ be a sequence of unknown loss vectors. The dynamics of OLO problem is stated in Algorithm 11. Given a time horizon $T \in \mathbb{N}$, the agent (or algorithm) aims to minimize the (loss-based) regret

$$\max_{p \in \mathbf{\Delta}_{[K]}} \sum_{t=1}^{T} (p_t - p)^\top \ell_t, \tag{17}$$

where $p_t$ is the action taken by the agent at time step $t$, $p$ is some fixed baseline action in $\mathbf{\Delta}_{[K]}$ and the maximum operator indicates the agent is competing with the best fixed baseline. The FTRL algorithm minimizes (17) by taking the action $p_t = \text{argmin}_{p \in \mathbf{\Delta}_{[K]}} \Phi_t(p)$ at time step $t$, where $\Phi_t : \mathbf{\Delta}_{[K]} \to \mathbb{R}$ is defined as

$$\Phi_t(x) := F_t(x) + \sum_{s=1}^{t-1} x^\top \ell_s$$

and $F_t : \mathbf{\Delta}_{[K]} \to \mathbb{R}, \forall t \in \mathbb{N}$, are some convex functions.

In the following, we illustrate the connection between FTRL and Algorithm 5 (or Algorithm 10). We let $\ell_t = \widehat{Z}_t := \left(\widehat{Z}_{1,t} \ldots, \widehat{Z}_{K,t}\right)^\top$, $F_t(x) = F(x)/\eta_t$ with $F(x) : \Delta_{[K]} \to \mathbb{R}$ and $F(x) := \sum_{i=1}^{K}(x_i \log x_i - x_i) + \log K + 1$. Furthermore, some calculation indicates the action taken by FTRL is exactly $p_t$ as in (15), i.e.,

$$p_t = \underset{p \in \Delta_{[K]}}{\arg\min} \, \Phi_t(p).$$

Therefore, Algorithm 5 is indeed an instantiation of FTRL in terms of the algorithm design.

**Under our problem setup**, the difference lies at the target of the algorithm. Instead of minimizing the corresponding regret

$$\max_{p \in \Delta_{[K]}} \mathbb{E}\left[ \sum_{t=1}^{T} (p_t - p)^\top \widehat{Z}_t \right],$$

we aim at minimizing the (loss-based) regret defined on two different time scales

$$\widetilde{\mathrm{Reg}}(\mathtt{ALG}) := \mathbb{E}\left[ \sum_{t=1}^{\mathrm{ST}(\mathtt{ALG})} p_t^\top \widehat{Z}_t \right] - \min_{i \in [K]} \mathbb{E}\left[ \sum_{t=1}^{\mathrm{ST}(\mathtt{ALG}_i)} e_i^\top \widehat{Z}_t \right], \tag{18}$$

where $e_i \in \mathbb{R}^K$ is an one-hot vector with the $i^{th}$ coordinate being 1, and the expectation is taken w.r.t. the internal randomness within $\mathtt{ALG}$ and $\widehat{Z}_t$. We highlight again that $\mathrm{ST}(\mathtt{ALG})$ is random whereas $\mathrm{ST}(\mathtt{ALG}_i)$ is fixed under the greedy decoding strategy.

**Step 2: Transformation of the stopping time regret.**

The stopping time regret (5) and $\widetilde{\mathrm{Reg}}(\mathtt{ALG})$ in (18) may look different at first sight. We demonstrate that these two notions of regret can be transformed into one another up to some constant factors.

We firstly simplify $\widetilde{\mathrm{Reg}}(\mathtt{ALG})$. Let $z_{i,t} = 1 - (y_{i,t} - 1)/L$. According to the definition of $p_t$ in (15) and $\widehat{Z}_t$ in (16),

$$\sum_{t=1}^{\mathrm{ST}(\mathtt{ALG})} p_t^\top \widehat{Z}_t = \sum_{t=1}^{\mathrm{ST}(\mathtt{ALG})} \frac{L + 1 - y_{i,t}}{L} = \frac{L+1}{L} \cdot \mathrm{ST}(\mathtt{ALG}) - \frac{\mathrm{len}(\mathrm{pt}_{\mathrm{ST}(\mathtt{ALG})})}{L}.$$

Furthermore, note that $\widehat{Z}_t$ is an unbiased estimator for $z_t$ and $\mathrm{ST}(\mathtt{ALG}_i)$ is a fixed real number,

$$\mathbb{E}\left[ \sum_{t=1}^{\mathrm{ST}(\mathtt{ALG}_i)} e_i^\top \widehat{Z}_t \right] = \sum_{t=1}^{\mathrm{ST}(\mathtt{ALG}_i)} z_{i,t} = \frac{L+1}{L} \cdot \mathrm{ST}(\mathtt{ALG}_i) - \frac{\mathrm{len}(\mathrm{pt}_{\mathrm{ST}(\mathtt{ALG}_i)})}{L}. \tag{19}$$

Hence, $\widetilde{\mathrm{Reg}}(\mathtt{ALG})$ is simplified as

$$\widetilde{\mathrm{Reg}}(\mathtt{ALG}) = \mathbb{E}\left[ \frac{L+1}{L} \cdot \mathrm{ST}(\mathtt{ALG}) - \frac{\mathrm{len}(\mathrm{pt}_{\mathrm{ST}(\mathtt{ALG})})}{L} \right] - \min_{i \in [K]}\left( \frac{L+1}{L} \cdot \mathrm{ST}(\mathtt{ALG}_i) - \frac{\mathrm{len}(\mathrm{pt}_{\mathrm{ST}(\mathtt{ALG}_i)})}{L} \right)$$

$$= \frac{L+1}{L} \cdot \left( \mathbb{E}[\mathrm{ST}(\mathtt{ALG})] - \min_{i \in [K]} \mathrm{ST}(\mathtt{ALG}_i) \right)$$

where we adopt the property of speculative decoding (2) in the last equality. This indicates that

$$\widetilde{\mathrm{Reg}}(\mathtt{ALG}) = \frac{L+1}{L} \cdot \mathrm{Reg}(\mathtt{ALG}). \tag{20}$$

**Step 3: Regret decomposition.**

Based on the previous two steps, we now upper bound $\widetilde{\mathrm{Reg}}(\mathtt{ALG})$. This notion of regret distinguishes from the standard regret analysis due to the difference in the time scales $\mathrm{ST}(\mathtt{ALG})$ and $\mathrm{ST}(\mathtt{ALG}_i)$.

For simplicity, we define the Bregman divergence induced by convex function $f : \mathbf{\Delta}_{[K]} \to \mathbb{R}^+$ as $\mathrm{D}_f(\cdot, \cdot) : \mathbf{\Delta}_{[K]} \times \mathbf{\Delta}_{[K]} \to \mathbb{R}$ with $\mathrm{D}_f(a, b) := f(a) - f(b) - (a - b)^\top \nabla f(b)$. Fix $i \in [K]$, we now decompose this empirical regret w.r.t. $i$.

$$\sum_{t=1}^{\mathrm{ST(ALG)}} p_t^\top \widehat{Z}_t - \sum_{t=1}^{\mathrm{ST(ALG}_i)} e_i^\top \widehat{Z}_t.$$

$$= \sum_{t=1}^{\mathrm{ST(ALG)}} \left( (p_t - p_{t+1})^\top \widehat{Z}_t \right) + \sum_{t=1}^{\mathrm{ST(ALG)}} p_{t+1}^\top \widehat{Z}_t - \sum_{t=1}^{\mathrm{ST(ALG}_i)} e_i^\top \widehat{Z}_t$$

$$= \sum_{t=1}^{\mathrm{ST(ALG)}} \left( (p_t - p_{t+1})^\top \widehat{Z}_t \right) + \sum_{t=1}^{\mathrm{ST(ALG)}} \left( \Phi_{t+1}(p_{t+1}) - F_{t+1}(p_{t+1}) - \Phi_t(p_{t+1}) + F_t(p_{t+1}) \right)$$

$$- \left( \sum_{t=1}^{\mathrm{ST(ALG)}} e_i^\top \widehat{Z}_t - \sum_{t=1}^{\mathrm{ST(ALG)}} e_i^\top \widehat{Z}_t + \sum_{t=1}^{\mathrm{ST(ALG}_i)} e_i^\top \widehat{Z}_t \right)$$

$$= \sum_{t=1}^{\mathrm{ST(ALG)}} \left( (p_t - p_{t+1})^\top \widehat{Z}_t \right) + \sum_{t=0}^{\mathrm{ST(ALG)}-1} \left( \Phi_{t+1}(p_{t+1}) - \Phi_{t+1}(p_{t+2}) \right)$$

$$+ \sum_{t=1}^{\mathrm{ST(ALG)}} \left( F_t(p_{t+1}) - F_{t+1}(p_{t+1}) \right) + F_{\mathrm{ST(ALG)}+1}(e_i) - F_1(p_1)$$

$$+ \Phi_{\mathrm{ST(ALG)}+1}(p_{\mathrm{ST(ALG)}+1}) - \Phi_{\mathrm{ST(ALG)}+1}(e_i) + \left( \sum_{t=1}^{\mathrm{ST(ALG)}} e_i^\top \widehat{Z}_t - \sum_{t=1}^{\mathrm{ST(ALG}_i)} e_i^\top \widehat{Z}_t \right)$$

$$\leq \underbrace{\sum_{t=1}^{\mathrm{ST(ALG)}} \left( (p_t - p_{t+1})^\top \widehat{Z}_t - \frac{\mathrm{D}_F(p_{t+1}, p_t)}{\eta_t} \right)}_{(\square)} + \underbrace{F_{\mathrm{ST(ALG)}+1}(e_i) - F_1(p_1)}_{(\diamond)}$$

$$+ \underbrace{\sum_{t=1}^{\mathrm{ST(ALG)}} \left( F_t(p_{t+1}) - F_{t+1}(p_{t+1}) \right)}_{(\dagger)} + \underbrace{\Phi_{\mathrm{ST(ALG)}+1}(p_{\mathrm{ST(ALG)}+1}) - \Phi_{\mathrm{ST(ALG}_i)+1}(e_i)}_{(\ddagger)}$$

$$+ \underbrace{\left( \sum_{t=1}^{\mathrm{ST(ALG)}} e_i^\top \widehat{Z}_t - \sum_{t=1}^{\mathrm{ST(ALG}_i)} e_i^\top \widehat{Z}_t \right)}_{(\P)} \tag{21}$$

where the last inequality adopts the fact that $\mathrm{D}_{\Phi_t}(a, b) = \mathrm{D}_{F_t}(a, b) = \mathrm{D}_F(a, b)/\eta_t$ and the inequality

$$\Phi_t(p_t) - \Phi_t(p_{t+1}) = -\mathrm{D}_{\Phi_t}(p_{t+1}, p_t) - (p_{t+1} - p_t)^\top \nabla \Phi_t(p_t) \leq -\mathrm{D}_{\Phi_t}(p_{t+1}, p_t).$$

Here $(p_{t+1} - p_t)^\top \nabla \Phi_t(p_t) \leq 0$ results from the choice of $p_t = \mathrm{argmin}_{p \in \mathbf{\Delta}_{[K]}} \Phi_t(p)$ and the first-order optimization condition.

**Step 4: Upper bound each term in the decomposed regret.**

In this step, we upper bound each term in (21). We comment that $(\square)$ and $(\P)$ require us to attend to the randomness in $\mathrm{ST(ALG)}$ and the different time indeces. This issue will not be encountered in the conventional scenario where $\mathrm{ST(ALG)} = \mathrm{ST(ALG}_i)$.

• Upper bound $(\square)$: we will show that $\mathbb{E}[(\square)] \leq \mathbb{E}[K \sum_{t=1}^{\mathrm{ST(ALG)}} \eta_t/2]$ almost surely.

Recall the definition of $\widehat{Z}_t$ in (6), $\widehat{Z}_t$ only has a positive value at the $I_t$ coordinate. We divide the problem into two cases: (1) $p_{t,I_t} - p_{t+1,I_t} < 0$, and (2) $p_{t,I_t} - p_{t+1,I_t} \geq 0$.

Case (1): $p_{t,I_t} - p_{t+1,I_t} < 0$. Because the Bregman divergence is always non-negative, hence,

$$(\square) \leq (p_{t,I_t} - p_{t+1,I_t}) \cdot \widehat{Z}_{I_t,t} - 0 \leq 0 \leq \frac{\eta_t}{2p_{t,I_t}}.$$

Case (2): $p_{t,I_t} - p_{t+1,I_t} \geq 0$. Note that $F(x)$ is a Legendre function on $\boldsymbol{\Delta}_{[K]}$. By invoking Lemma F.8,

$$(p_t - p_{t+1})^\top \widehat{Z}_t - \frac{\mathrm{D}_F(p_{t+1}, p_t)}{\eta_t} \leq \frac{\eta_t}{2} \|\widehat{Z}_t\|^2_{H_t^{-1}},$$

where $H_t = \nabla^2 F(q_t)$ and $q_t = \alpha \cdot p_t + (1-\alpha) \cdot p_{t+1}$ for some $\alpha \in [0,1]$. Furthermore, $\nabla^2 F(q_t)$ is a $K \times K$ diagonal matrix with $\left(\nabla^2 F(q_t)\right)_{i,i} = 1/q_{t,i}, \forall i \in [K]$. Therefore,

$$\frac{\eta_t}{2} \|\widehat{Z}_t\|^2_{H_t^{-1}} = \frac{\eta_t}{2} \cdot \frac{z^2_{I_t,t}}{p^2_{t,I_t}} \cdot q_{t,I_t} \leq \frac{\eta_t}{2} \cdot \frac{1}{p^2_{t,I_t}} \cdot p_{t,I_t} = \frac{\eta_t}{2p_{t,I_t}}.$$

To conclude the two cases, it holds almost surely that

$$(\square) \leq \sum_{t=1}^{\mathrm{ST}(\mathtt{ALG})} \frac{\eta_t}{2p_{t,I_t}}.$$

Lastly, by adopting Lemma D.1 which is proved in Appendix E.1, we obtain

$$\mathbb{E}[(\square)] \leq \frac{K}{2} \cdot \mathbb{E}\left[\sum_{t=1}^{\mathrm{ST}(\mathtt{ALG})} \eta_t\right].$$

**Lemma D.1.** *Under Assumption 4.2, consider the learning rates $\eta_t$ defined in Algorithm 5 (or Algorithm 10), it holds that*

$$\mathbb{E}\left[\sum_{t=1}^{\mathrm{ST}(\mathtt{ALG})} \frac{\eta_t}{p_{t,I_t}}\right] = K \cdot \mathbb{E}\left[\sum_{t=1}^{\mathrm{ST}(\mathtt{ALG})} \eta_t\right].$$

• Upper bound ($\diamond$): Because $F(x)$ is non-negative on $\boldsymbol{\Delta}_{[K]}$ and $F(e_i) = \log K$, hence,

$$(\diamond) \leq \frac{F(e_i)}{\eta_{\mathrm{ST}(\mathtt{ALG}+1)}} = \frac{\log K}{\eta_{\mathrm{ST}(\mathtt{ALG})+1}} = \frac{\log K}{\eta_{\mathrm{ST}(\mathtt{ALG})}}.$$

where we manually set $\eta_{\mathrm{ST}(\mathtt{ALG})+1} = \eta_{\mathrm{ST}(\mathtt{ALG})}$.

• Upper bound ($\dagger$): Recall that $F(x) = \sum_{i=1}^K (x_i \log x_i - x_i) + \log K + 1$, so $F(x)$ is non-negative for any $x \in \boldsymbol{\Delta}_{[K]}$. Additionally, $\eta_t = \sqrt{\log K/(K \cdot t)}, t \in [\mathrm{ST}(\mathtt{ALG})]$ is a decreasing sequence. Therefore,

$$(\dagger) = \sum_{t=1}^{\mathrm{ST}(\mathtt{ALG})} \left(\frac{F(p_{t+1})}{\eta_t} - \frac{F(p_{t+1})}{\eta_{t+1}}\right) \leq 0, \quad a.s.$$

• Upper bound ($\ddagger$): Since $p_{\mathrm{ST}(\mathtt{ALG})+1}$ is the minimizer of $\Phi_{\mathrm{ST}(\mathtt{ALG})+1}(p)$, we have ($\ddagger$) $\leq 0$.

• Upper bound ($\P$): Under Assumption 5.2, $\mathrm{ST}(\mathtt{ALG}) \geq \mathrm{ST}(\mathtt{ALG}_1)$ almost surely. We further prove that $\mathbb{E}[(\P)] \leq \mathbb{E}[\mathrm{ST}(\mathtt{ALG})] - \mathrm{ST}(\mathtt{ALG}_i)$ for $i = i^*$, which we summarize in the following lemma with proof postponed to Appendix E.1.

**Lemma D.2.** *Under Assumption 4.2 and 5.2, consider $\widehat{Z}_t$ defined in Algorithm 5, it holds that*

$$\mathbb{E}\left[\sum_{t=\mathrm{ST}(\mathtt{ALG}_{i^*})+1}^{\mathrm{ST}(\mathtt{ALG})} e_{i^*}^\top \widehat{Z}_t\right] \leq \mathbb{E}[\mathrm{ST}(\mathtt{ALG})] - \mathrm{ST}(\mathtt{ALG}_{i^*}).$$

**Step 5: Conclusion of the stopping time regret.** Aggregating the upper bounds for each terms in (21) and taking expectation,

$$\widetilde{\mathrm{Reg}}(\mathtt{ALG}) \leq \frac{K}{2}\mathbb{E}\left[\sum_{t=1}^{\mathrm{ST}(\mathtt{ALG})}\eta_t\right] + \mathbb{E}\left[\frac{\log K}{\eta_{\mathrm{ST}(\mathtt{ALG})}}\right] + \mathbb{E}[\mathrm{ST}(\mathtt{ALG})] - \mathrm{ST}(\mathtt{ALG}_{i^*}).$$

By substituting the learning rate values into the equation,

$$\widetilde{\mathrm{Reg}}(\mathtt{ALG}) \leq 2 \cdot \mathbb{E}\left[\sqrt{\mathrm{ST}(\mathtt{ALG}) \cdot K \log K}\right] + \mathbb{E}[\mathrm{ST}(\mathtt{ALG})] - \mathrm{ST}(\mathtt{ALG}_{i^*})$$

$$\leq 2 \cdot \sqrt{\mathbb{E}\left[\mathrm{ST}(\mathtt{ALG})\right] \cdot K \log K} + \mathbb{E}[\mathrm{ST}(\mathtt{ALG})] - \mathrm{ST}(\mathtt{ALG}_{i^*})$$

$$\leq 2 \cdot \sqrt{\mathrm{len}(\mathrm{pt}_{\tau_c}) \cdot K \log K} + \mathbb{E}[\mathrm{ST}(\mathtt{ALG})] - \mathrm{ST}(\mathtt{ALG}_{i^*}).$$

where the second inequality adopts Jensen's inequality and the last equality holds due to $\mathrm{ST}(\mathtt{ALG}) \leq \mathrm{len}(\mathrm{pt}_{\tau_c})$ almost surely as in (3). According to the regret transformation in (20),

$$\mathrm{Reg}(\mathtt{ALG}) \leq 2L \cdot \sqrt{\mathbb{E}\left[\mathrm{ST}(\mathtt{ALG})\right] \cdot K \log K} \leq 2L \cdot \sqrt{\mathrm{len}(\mathrm{pt}_{\tau_c}) \cdot K \log K}. \tag{22}$$

Furthermore, by solving the quadratic function in terms of $\mathrm{Reg}(\mathtt{ALG})$, i.e.,

$$\mathrm{Reg}(\mathtt{ALG}) \leq 2L \cdot \sqrt{\left(\mathrm{Reg}(\mathtt{ALG}) + \mathrm{ST}(\mathtt{ALG}_{i^*})\right) \cdot K \log K},$$

we obtain

$$\mathrm{Reg}(\mathtt{ALG}) \leq 4L^2 \cdot K \log K + 2L \cdot \sqrt{\min_{i \in [K]} \mathrm{ST}(\mathtt{ALG}_i) \cdot K \log K}. \tag{23}$$

Aggregating (22) and (23) concludes the proof of this theorem. $\qquad\square$

*Remark* D.3 (Sampling Decoding under Adversarial Mean Values). Since the tokens can be regarded as fixed given the initial prompt and the hyperparameter configurations under the greedy decoding strategy, we consider the greedy decoding strategy under the adversarial mean values assumption (Assumption 5.1). If one wishes to consider the sampling decoding strategy, the proof of Theorem 5.3 can be adapted to it. Specifically, this switch of decoding strategy mainly influences (19), the proofs of Lemma D.1 and Lemma D.2. We can depend on Doob's Optional Stopping Theorem (Lemma F.1) to solve this problem, just like what we have done to prove (28) and replacing the condition (1) therein by $\mathbb{E}[\mathrm{ST}(\mathtt{ALG})] \leq \mathbb{E}[\mathrm{len}(\mathrm{pt}_{\tau_c})] < \infty$. The rest of the proof can go through in a similar manner. In the end, we can arrive at a similar result, i.e.,

$$\mathrm{Reg}(\mathtt{ALG}, \mathrm{pt}, \nu) \leq 2L \cdot \min\left\{2LK \log K + \sqrt{\min_{i \in [K]} \mathbb{E}[\mathrm{ST}(\mathtt{ALG}_i)]K \log K}, \sqrt{\mathbb{E}[\mathrm{len}(\mathrm{pt}_{\tau_c})]K \log K}\right\}.$$

### D.4. Proof of Theorem 4.4

Under the greedy decoding strategy, the problem is alleviated in two aspects. **Firstly**, given the target model $P$ and the initial prompt $\mathrm{pt}$, the total length $\mathrm{len}(\mathrm{pt}_{\tau_c})$ is (potentially) determined. While the total length is determined, it is worth noting that the number of accepted tokens at each round (Line 5 in Algorithm 3) is *still random*. **Additionally**, under the dynamics presented in Algorithm 3 and given the history $\mathcal{H}_t$, there is a one-to-one mapping between the accepted tokens $X_{I_t,t}$ and its length $Y_{I_t,t}$.

Since the lower bound is established in terms of a class of algorithms over a set of initial prompts, we adopt $\mathcal{X}^*_{initial}$ to denote the set of initial prompts and adopt $\mathcal{S}_{all}$ to denote the set of all hyperparameter specifications that can be selected to constitute $\mathcal{S}$. To further ease the problem, we augment Assumption 4.1.

**Assumption D.4.** We assume that
• Given any bandit configuration $\nu = (P, \mathcal{S} = \{S_i\}_{i \in [K]}, L)$ and initial prompt $\mathrm{pt} \in \mathcal{X}^*_{initial}$, conditional on the history $\mathcal{H}_{t-1}$ and the selected arm $I_t$ at round $t$, the distribution of the length of the accepted tokens $\mathbb{P}(\,\cdot\, | \mathrm{pt}, \mathcal{H}_{t-1}, I_t = i) = P_{S_i}(\,\cdot\,), \forall i \in [K]$.
• For any two hyperparameter specifications $S, S' \in \mathcal{S}_{all}$, we have $\mathrm{KL}(P_S, P_{S'}) < \infty$.

We consider the class of arm selection algorithms which are *non-anticipatory* and *consistent*.

**Definition D.5** (Non-anticipartory Algorithm). An arm selection algorithm ALG is non-anticipatory if $\text{ALG}(\cdot \mid \mathcal{H}_t) \in \sigma(\mathcal{H}_t), \; \forall t \in \mathbb{N}$.

**Definition D.6** (Consistent Algoithm). An arm selection algorithm ALG is consistent over a class of bandit configurations $\Lambda$ and prompt set $\mathcal{X}_{initial}^*$ if for all $\nu \in \Lambda$ and any sequence of initial prompts $(\text{pt}^m)_{m=1}^\infty \subset \mathcal{X}_{initial}^*$ with $\text{len}(\text{pt}_{\tau_c}^m) \to \infty, m \to \infty$, and for all $a \in (0,1]$,

$$\lim_{m\to\infty} \frac{\text{Reg}(\text{ALG}, \text{pt}^m, \nu)}{\text{len}(\text{pt}_{\tau_c}^m)^a} = 0.$$

**Theorem 4.4** (Lower Bound). *Given any sequence of initial prompts* $(\text{pt}^m)_{m=1}^\infty \subset \mathcal{X}_{\text{init}}^*$ *with* $\text{len}(\text{pt}_{\tau_c}^m) \to \infty, m \to \infty$ *and a bandit configuration* $\nu = (P, \mathcal{S} = \{S_i\}_{i \in [K]}, L)$, *under Assumption D.4, the greedy decoding strategy and the dynamics represented in Algorithm 3, for any non-anticipatory and consistent arm selection algorithm* ALG, *the expected regret satisfies*

$$\liminf_{m\to\infty} \frac{\text{Reg}(\text{ALG}, \text{pt}, \nu)}{\log(\text{len}(\text{pt}_{\tau_c}^m))} \geq \sum_{i \neq i^*} \frac{\Delta_i}{\mu_{i^*}} \cdot \frac{1}{\text{kl}_i},$$

*where* $\text{kl}_i := \inf_{S \in \mathcal{S}} \{\text{KL}(P_{S_i}, P_S) : \mathbb{E}_{X \sim P_S}[X] > \mu_{i^*}\}$.

*Proof of Theorem 4.4.* The proof consists of three steps:

- Divergence decomposition: similar to the reward decomposition in the upper bound proof, the divergence decomposition cannot be done as the time horizon $\text{ST}(\text{ALG})$ is a random stopping time. We tackle this problem in the first step.

- Lower bound on $\mathbb{E}_\nu[n_{i,\text{ST}(\text{ALG})}]$: we adapt the standard trick to lower bound the expected number of times arm $i$ has been chosen.

- Conclusion of the proof.

**Step 1: Divergence decomposition.** The divergence decomposition suffers from the same issue as the reward decomposition, i.e., the stopping time $\text{ST}(\text{ALG})$ depends on the history. We adopt the same trick as in the reward decomposition step to overcome this issue and the result is summarized in Lemma D.7 whose proof is postponed to App. E.

**Lemma D.7.** *Under Assumption D.4, given two bandit configurations* $\nu = (P, \mathcal{S} = \{S_i\}_{i=1}^K, L)$ *and* $\nu' = (P, \mathcal{S}' = \{S_i'\}_{i=1}^K, L)$ *which only differ in the hyperparameter specifications, for any* $\text{pt} \in \mathcal{X}_{initial}^*$ *and algorithm* ALG,

$$\text{KL}(\mathbb{P}_{\text{ALG}, \text{pt}, \nu}, \mathbb{P}_{\text{ALG}, \text{pt}, \nu'}) = \sum_{i=1}^K \mathbb{E}_{\text{ALG}, \text{pt}, \nu}[n_{i,\text{ST}(\text{ALG})}] \text{KL}(P_{S_i}, P_{S_i'})$$

*where* $\mathbb{P}_{\text{ALG}, \text{pt}, \nu}$ *(resp.* $\mathbb{P}_{\text{ALG}, \text{pt}, \nu}$*) is the probability measure induced by* $(\text{ALG}, \text{pt}, \nu)$ *(resp.* $(\text{ALG}, \text{pt}, \nu)$*) defined on the* $\sigma$*-algebra* $\{\sigma(\mathcal{H}_t)\}_{t=1}^\infty$.

**Step 2: Establishment for the lower bound of** $\mathbb{E}_\nu[n_{i,\text{ST}(\text{ALG})}]$. Given algorithm ALG, bandit configuration $\nu \in \Lambda$, prompt $\text{pt} \in (\text{pt}^m)_{m=1}^\infty$ and any $\varepsilon > 0$, construct $K - 1$ alternative bandit configurations $\nu_i = (P, \mathcal{S}_i = \{S_{i,j}\}_{j=1}^K, L)$ for $i \neq i^*$ with

$$S_{i,j} = S_j \cdot \mathbb{1}\{j \neq i\} + S_i' \cdot \mathbb{1}\{j = i\},$$

where $S_i'$ in $\nu_i$ satisfies that its mean $\mu_{S_i'} > \mu_{i^*}$ with $\text{KL}(P_{S_i}, P_{S_i'}) \leq \text{kl}_i + \varepsilon$. In other words, under bandit configuration $\nu_i$, only $S_i$ changes into $S_i'$ and arm $i$ becomes the best arm. As the bandit configurations only differ in the hyperparameter selection, we adopt the shorthand notation $\mathbb{P}_\nu, \mathbb{E}_\nu$ and $\text{Reg}(\nu)$ for $\mathbb{P}_{\text{ALG}, \text{pt}, \nu}, \mathbb{E}_{\text{ALG}, \text{pt}, \nu}$ and $\text{Reg}(\text{ALG}, \text{pt}, \nu)$ respectively when there is no risk of confusion.

According to Lemma D.7, for any pt $\in (\text{pt}^m)_{m=1}^\infty$,

$$\text{KL}(\mathbb{P}_\nu, \mathbb{P}_{\nu_i}) = \mathbb{E}_\nu[n_{i,\text{ST(ALG)}}] \cdot \text{KL}(P_{S_i}, P_{S_i'}) \le \mathbb{E}_\nu[n_{i,\text{ST(ALG)}}] \cdot (\text{kl}_i + \varepsilon).$$

By Lemma F.6, with $A_i = \{n_{i,\text{ST(ALG)}} > \text{ST(ALG)}/2\}$,

$$\mathbb{P}_\nu[A_i] + \mathbb{P}_{\nu_i}[A_i^c] \ge \frac{1}{2} \exp\left( -\mathbb{E}_\nu[n_{i,\text{ST(ALG)}}] \cdot \text{KL}(P_{S_i}, P_{S_i'}) \right) \tag{24}$$
$$\ge \frac{1}{2} \exp\left( -\mathbb{E}_\nu[n_{i,\text{ST(ALG)}}] \cdot (\text{kl}_i + \varepsilon) \right)$$

Thus, by adopting (14), the expected reward under $\nu$ can be lower bounded as

$$\text{Reg}(\nu) \ge \frac{\Delta_i}{\mu_{i^*}} \cdot \mathbb{E}_\nu[n_{i,\text{ST(ALG)}}] \ge \frac{\Delta_i}{\mu_{i^*}} \cdot \mathbb{E}_\nu[n_{i,\text{ST(ALG)}} \mid A_i] \cdot \mathbb{P}_\nu[A_i] \tag{25}$$
$$\ge \frac{\Delta_i}{\mu_{i^*}} \cdot \frac{1}{2} \cdot \mathbb{E}_\nu[\text{ST(ALG)} \mid A_i] \cdot \mathbb{P}_\nu[A_i] \ge \frac{\Delta_i}{\mu_{i^*}} \cdot \frac{\text{len}(\text{pt}_{\tau_c})}{2(L+1)} \cdot \mathbb{P}_\nu[A_i].$$

Similarly, under the alternative bandit configuration $\nu_i$,

$$\text{Reg}(\nu_i) \ge \frac{\mu_{S_i'} - \mu_{i^*}}{\mu_{S_i'}} \cdot \mathbb{E}_{\nu_i}[n_{i,\text{ST(ALG)}}] \ge \frac{\mu_{S_i'} - \mu_{i^*}}{\mu_{S_i'}} \cdot \frac{\text{len}(\text{pt}_{\tau_c})}{2(L+1)} \cdot \mathbb{P}_{\nu_i}[A_i^c]. \tag{26}$$

Combining (24), (25) and (26),

$$\text{Reg}(\nu) + \text{Reg}(\nu_i)$$
$$\ge \min\left\{ \frac{\Delta_i}{\mu_{i^*}}, \frac{\mu_{S_i'} - \mu_{i^*}}{\mu_{S_i'}} \right\} \cdot \frac{\text{len}(\text{pt}_{\tau_c})}{2(L+1)} (\mathbb{P}_\nu[A_i] + \mathbb{P}_{\nu_i}[A_i^c])$$
$$\ge \min\left\{ \frac{\Delta_i}{\mu_{i^*}}, \frac{\mu_{S_i'} - \mu_{i^*}}{\mu_{S_i'}} \right\} \cdot \frac{\text{len}(\text{pt}_{\tau_c})}{4(L+1)} \cdot \exp\left( -\mathbb{E}_\nu[n_{i,\text{ST(ALG)}}] \cdot (\text{kl}_i + \varepsilon) \right)$$

which holds for any pt $\in (\text{pt}_m)_{m=1}^\infty$. Rearranging the terms, we have

$$\liminf_{m\to\infty} \frac{\mathbb{E}_\nu[n_{i,\text{ST(ALG)}}]}{\log(\text{len}(\text{pt}_{\tau_c}^m))}$$
$$\ge \frac{1}{\text{kl}_i + \varepsilon} + \liminf_{m\to\infty} \frac{\log\left( \min\left\{ \frac{\Delta_i}{\mu_{i^*}}, \frac{\mu_{S_i'} - \mu_{i^*}}{\mu_{S_i'}} \right\} \right) - \log\left( 4(L+1) \right) - \log(\text{Reg}(\nu) + \text{Reg}(\nu_i))}{(\text{kl}_i + \varepsilon) \cdot \log(\text{len}(\text{pt}_{\tau_c}^m))}$$
$$= \frac{1}{\text{kl}_i + \varepsilon}$$

where the last equality follows from the definition of consistent algorithm. Because $\varepsilon > 0$ is arbitrarily chosen, by sending $\varepsilon \to 0$, we obtain the lower bound for $\mathbb{E}_\nu[n_{i,\text{ST(ALG)}}]$

$$\liminf_{m\to\infty} \frac{\mathbb{E}_\nu[n_{i,\text{ST(ALG)}}]}{\log(\text{len}(\text{pt}_{\tau_c}^m))} \ge \frac{1}{\text{kl}_i}. \tag{27}$$

**Step 3: Conclusion of the proof.** Aggregating (14) and (27),

$$\liminf_{m\to\infty} \frac{\text{Reg}(\text{ALG}, \text{pt}^m, \nu)}{\log(\text{len}(\text{pt}_{\tau_c}^m))} \ge \sum_{i\ne i^*} \frac{\Delta_i}{\mu_{i^*}} \cdot \frac{1}{\text{kl}_i}.$$

This concludes the proof. $\qquad\square$

# E. Supporting Propositions

### E.1. Supporting Lemmas for Theorem 5.3

**Lemma D.1.** *Under Assumption 4.2, consider the learning rates $\eta_t$ defined in Algorithm 5 (or Algorithm 10), it holds that*

$$\mathbb{E}\left[\sum_{t=1}^{\text{ST}(\text{ALG})} \frac{\eta_t}{p_{t,I_t}}\right] = K \cdot \mathbb{E}\left[\sum_{t=1}^{\text{ST}(\text{ALG})} \eta_t\right].$$

*Proof of Lemma D.1.* Similar to the proof of Lemma D.2, we adopt Doob's Optional Stopping Theorem (Lemma F.1) to show

$$\mathbb{E}\left[\sum_{t=1}^{\text{ST}(\text{ALG})} \frac{\eta_t}{p_{t,I_t}} - K \cdot \eta_t\right] = 0. \tag{28}$$

According to condition $(b)$ in Lemma F.1, we only need to show that (1) $\mathbb{E}[\text{ST}(\text{ALG})] < \infty$, and (2) there exists $c \in \mathbb{R}$ such that for all $t < \text{ST}(\text{ALG})$, $\mathbb{E}\left[\left|\eta_t/p_{t,I_t} - K \cdot \eta_t\right| \middle| \mathcal{H}_{t-1}\right] < c$.

Condition $(1)$: Given Assumption 4.2, it holds that $\text{len}(\text{pt}_{\tau_c}) < \infty$ under the greedy decoding strategy. Therefore, we have $\overline{\mathbb{E}[\text{ST}(\text{ALG})]} \leq \text{len}(\text{pt}_{\tau_c}) < \infty$.

Condition $(2)$: Note that

$$\mathbb{E}\left[\left|\frac{\eta_t}{p_{t,I_t}} - K \cdot \eta_t\right| \middle| \mathcal{H}_{t-1}\right] \leq \mathbb{E}\left[\frac{\eta_t}{p_{t,I_t}} \middle| \mathcal{H}_{t-1}\right] + K \cdot \eta_t \leq \mathbb{E}\left[\sum_{i=1}^{K} \frac{\eta_t}{p_{t,i}} \mathbb{1}\{I_t = i\} \middle| \mathcal{H}_{t-1}\right] + K \cdot \eta_t$$

$$\leq 2K \cdot \eta_t \leq 2\sqrt{K \log K}.$$

Therefore, Condition $(2)$ is satisfied and (28) is established.

**In addition**, by using Assumption 4.2,

$$\mathbb{E}\left[\left|\sum_{t=1}^{\text{ST}(\text{ALG})} K \cdot \eta_t\right|\right] \leq \mathbb{E}\left[\text{ST}(\text{ALG})\right]\sqrt{K \log K} < \infty.$$

So $\mathbb{E}\left[\sum_{t=1}^{\text{ST}(\text{ALG})} \eta_t\right]$ exists. Lastly,

$$\mathbb{E}\left[\sum_{t=1}^{\text{ST}(\text{ALG})} \frac{\eta_t}{p_{t,I_t}}\right] = \mathbb{E}\left[\sum_{t=1}^{\text{ST}(\text{ALG})} \frac{\eta_t}{p_{t,I_t}} - K \cdot \eta_t\right] + \mathbb{E}\left[\sum_{t=1}^{\text{ST}(\text{ALG})} \eta_t\right]$$

which indicates $\mathbb{E}\left[\sum_{t=1}^{\text{ST}(\text{ALG})} \eta_t/p_{t,I_t}\right]$ exists.

**In conclusion**, by adding $K \cdot \mathbb{E}\left[\sum_{t=1}^{\text{ST}(\text{ALG})} \eta_t\right]$ on both sides of (28), the desired result is obtained. $\square$

**Lemma D.2.** *Under Assumption 4.2 and 5.2, consider $\widehat{Z}_t$ defined in Algorithm 5, it holds that*

$$\mathbb{E}\left[\sum_{t=\text{ST}(\text{ALG}_{i^*})+1}^{\text{ST}(\text{ALG})} e_{i^*}^\top \widehat{Z}_t\right] \leq \mathbb{E}[\text{ST}(\text{ALG})] - \text{ST}(\text{ALG}_{i^*}).$$

*Proof of Lemma D.2.* If the two expectations below exist and

$$\mathbb{E}\left[\sum_{t=1}^{\text{ST}(\text{ALG})} e_i^\top \widehat{Z}_t\right] = \mathbb{E}\left[\sum_{t=1}^{\text{ST}(\text{ALG})} z_{i,t}\right], \tag{29}$$

then it holds that

$$
\mathbb{E}\left[\sum_{t=1}^{\mathrm{ST}(\mathtt{ALG})} e_i^\top \widehat{Z}_t\right] - \mathbb{E}\left[\sum_{t=1}^{\mathrm{ST}(\mathtt{ALG}_i)} e_i^\top \widehat{Z}_t\right] = \mathbb{E}\left[\sum_{t=1}^{\mathrm{ST}(\mathtt{ALG})} z_{i,t}\right] - \sum_{t=1}^{\mathrm{ST}(\mathtt{ALG}_i)} z_{i,t}
$$

$$
\leq \mathbb{E}\left[\sum_{t=\mathrm{ST}(\mathtt{ALG}_i)+1}^{\mathrm{ST}(\mathtt{ALG})} z_{i,t}\right] \leq \mathbb{E}[\mathrm{ST}(\mathtt{ALG})] - \mathrm{ST}(\mathtt{ALG}_i).
$$

The desired result can be obtained.

Therefore, we aim to prove (29). Since both $\mathrm{ST}(\mathtt{ALG})$ and $\widehat{Z}_t$ are random, the obstacle is that we cannot directly take expectation of the summand. We will **firstly** prove

$$
\mathbb{E}\left[\sum_{t=1}^{\mathrm{ST}(\mathtt{ALG})} e_i^\top \widehat{Z}_t - z_{i,t}\right] = 0 \tag{30}
$$

by Doob's Optional Stopping Theorem (Lemma F.1). According to condition $(b)$ in Lemma F.1, we only need to show that (1) $\mathbb{E}[\mathrm{ST}(\mathtt{ALG})] < \infty$, and (2) there exists $c \in \mathbb{R}$ such that for all $t < \mathrm{ST}(\mathtt{ALG})$, $\mathbb{E}\big[|e_i^\top (\widehat{Z}_t - z_t)| \big| \mathcal{H}_{t-1}\big] < c$.

Condition (1) holds as shown in the proof Lemma D.1. We only need to show Condition (2). Note that $\mathbb{E}[\widehat{Z}] = z_t$,

$$
\mathbb{E}\big[|e_i^\top (\widehat{Z}_t - z_t)| \mid \mathcal{H}_{t-1}\big] \leq \mathbb{E}\Big[\mathbb{E}\big[e_i^\top \widehat{Z}_t \mid \mathcal{H}_{t-1}, I_t\big] \Big| \mathcal{H}_{t-1}\Big] + e_i^\top z_t = 2z_{i,t} \leq 2.
$$

Therefore, $\sum_{t=1}^{\mathrm{ST}(\mathtt{ALG})} e_i^\top \widehat{Z}_t - z_{i,t}$ is well-defined and (30) is established.

We **then** prove the two expectations exist. Note that all involved variables are positive, we only need to show

$$
\mathbb{E}\left[\sum_{t=1}^{\mathrm{ST}(\mathtt{ALG})} z_{i,t}\right] < \infty.
$$

This can be obtained by noticing that $z_{i,t} \in [0,1]$ for $t \in \mathbb{N}$,

$$
\mathbb{E}\left[\sum_{t=1}^{\mathrm{ST}(\mathtt{ALG})} z_{i,t}\right] \leq \mathbb{E}\big[\mathrm{ST}(\mathtt{ALG})\big] < \infty.
$$

Combining the above with Condition 2, we have

$$
\mathbb{E}\left[\sum_{t=1}^{\mathrm{ST}(\mathtt{ALG})} e_i^\top \widehat{Z}_t\right] = \mathbb{E}\left[\sum_{t=1}^{\mathrm{ST}(\mathtt{ALG})} e_i^\top \widehat{Z}_t - p^\top z_t\right] + \mathbb{E}\left[\sum_{t=1}^{\mathrm{ST}(\mathtt{ALG})} e_i^\top z_t\right]
$$

which means $\mathbb{E}\big[\sum_{t=1}^{\mathrm{ST}(\mathtt{ALG})} e_i^\top \widehat{Z}_t\big]$ exists.

**In conclusion**, by adding $\mathbb{E}\big[\sum_{t=1}^{\mathrm{ST}(\mathtt{ALG})} z_{i,t}\big]$ on both sides of (30),

$$
\mathbb{E}\left[\sum_{t=1}^{\mathrm{ST}(\mathtt{ALG})} e_i^\top \widehat{Z}_t\right] = \mathbb{E}\left[\sum_{t=1}^{\mathrm{ST}(\mathtt{ALG})} z_{i,t}\right].
$$

This finishes the proof of (29). $\qquad\square$

### E.2. Proof of Lemma D.7

**Lemma D.7.** *Under Assumption D.4, given two bandit configurations $\nu = (P, \mathcal{S} = \{S_i\}_{i=1}^K, L)$ and $\nu' = (P, \mathcal{S}' = \{S_i'\}_{i=1}^K, L)$ which only differ in the hyperparameter specifications, for any $\mathrm{pt} \in \mathcal{X}_{initial}^*$ and algorithm $\mathtt{ALG}$,*

$$
\mathrm{KL}(\mathbb{P}_{\mathtt{ALG},\mathrm{pt},\nu}, \mathbb{P}_{\mathtt{ALG},\mathrm{pt},\nu'}) = \sum_{i=1}^K \mathbb{E}_{\mathtt{ALG},\mathrm{pt},\nu}[n_{i,\mathrm{ST}(\mathtt{ALG})}]\mathrm{KL}(P_{S_i}, P_{S_i'})
$$

where $\mathbb{P}_{\mathtt{ALG},\mathrm{pt},\nu}$ (resp. $\mathbb{P}_{\mathtt{ALG},\mathrm{pt},\nu}$) is the probability measure induced by $(\mathtt{ALG}, \mathrm{pt}, \nu)$ (resp. $(\mathtt{ALG}, \mathrm{pt}, \nu)$) defined on the $\sigma$-algebra $\{\sigma(\mathcal{H}_t)\}_{t=1}^{\infty}$.

*Proof.* As the two bandit configurations only differ in the hyperparameter specifications, we adopt the abbreviated notation $\mathbb{P}_\nu$ and $\mathbb{P}_{\nu'}$ for the induced probability $\mathbb{P}_{\mathtt{ALG},\mathrm{pt},\nu}$ and $\mathbb{P}_{\mathtt{ALG},\mathrm{pt},\nu}$, respectively. We use $P_{\mathtt{ALG}}(\cdot)$ to denote the output distribution of the arm selection algorithm $\mathtt{ALG}$ in Line 4 in Algorithm 3.

With the bandit configuration $\nu = (P, \mathcal{S}, L)$, the probability of $\mathcal{H}_{\mathrm{ST}(\mathtt{ALG})}$ is

$$\mathbb{P}_\nu(\mathcal{H}_{\mathrm{ST}(\mathtt{ALG})}) = \prod_{t=1}^{\mathrm{ST}(\mathtt{ALG})} P_{\mathtt{ALG}}(I_t \mid \mathcal{H}_{t-1}, \mathrm{pt})\mathbb{P}(Y_{I_t,t} \mid \mathcal{H}_{t-1}, \mathrm{pt}, I_t) = \prod_{t=1}^{\mathrm{ST}(\mathtt{ALG})} P_{\mathtt{ALG}}(I_t \mid \mathcal{H}_{t-1}, \mathrm{pt}) P_{S_{I_t}}(Y_{I_t,t}).$$

Similarly, under the bandit configuration $\nu' = (P, \mathcal{S}', L)$,

$$\mathbb{P}_{\nu'}(\mathcal{H}_{\mathrm{ST}(\mathtt{ALG})}) = \prod_{t=1}^{\mathrm{ST}(\mathtt{ALG})} P_{\mathtt{ALG}}(I_t \mid \mathcal{H}_{t-1}, \mathrm{pt}) P_{S'_{I_t}}(Y_{I_t,t}).$$

Therefore, it holds that

$$\log \frac{\mathbb{P}_\nu(\mathcal{H}_{\mathrm{ST}(\mathtt{ALG})})}{\mathbb{P}_{\nu'}(\mathcal{H}_{\mathrm{ST}(\mathtt{ALG})})} = \sum_{t=1}^{\mathrm{ST}(\mathtt{ALG})} \log \frac{P_{S_{I_t}}(Y_{I_t,t})}{P_{S'_{I_t}}(Y_{I_t,t})}.$$

Because $\mathrm{KL}(P_{S_i}, P_{S'_i}) < \infty, \forall i \in [K]$ under Assumption D.4, the divergence between $\mathbb{P}_\nu$ and $\mathbb{P}_{\nu'}$ can be rewritten as

$$\mathrm{KL}(\mathbb{P}_\nu, \mathbb{P}_{\nu'}) = \mathbb{E}_\nu\left[\log \frac{\mathbb{P}_\nu(\mathcal{H}_{\mathrm{ST}(\mathtt{ALG})})}{\mathbb{P}_{\nu'}(\mathcal{H}_{\mathrm{ST}(\mathtt{ALG})})}\right] = \mathbb{E}_\nu\left[\sum_{t=1}^{\mathrm{ST}(\mathtt{ALG})} \log \frac{P_{S_{I_t}}(Y_{I_t,t})}{P_{S'_{I_t}}(Y_{I_t,t})}\right] < \infty.$$

We **then** prove that

$$\mathbb{E}_\nu\left[\sum_{t=1}^{\mathrm{ST}(\mathtt{ALG})} \log \frac{P_{S_{I_t}}(Y_{I_t,t})}{P_{S'_{I_t}}(Y_{I_t,t})}\right] = \sum_{i=1}^{K} \mathbb{E}_\nu[n_{i,\mathrm{ST}(\mathtt{ALG})}]\mathrm{KL}(P_{S_i}, P_{S'_i}). \tag{31}$$

The proof is composed by two arguments:

- **Argument 1**:

$$\mathbb{E}_\nu\left[\sum_{t=1}^{\mathrm{ST}(\mathtt{ALG})} \left(\log \frac{P_{S_{I_t}}(Y_{I_t,t})}{P_{S'_{I_t}}(Y_{I_t,t})} - \mathrm{KL}(P_{S_{I_t}}, P_{S'_{I_t}})\right)\right] = 0. \tag{32}$$

- **Argument 2**:

$$\mathbb{E}_\nu\left[\sum_{t=1}^{\mathrm{ST}(\mathtt{ALG})} \mathrm{KL}(P_{S_{I_t}}, P_{S'_{I_t}})\right] = \sum_{i=1}^{K} \mathbb{E}_\nu[n_{i,\mathrm{ST}(\mathtt{ALG})}]\mathrm{KL}(P_{S_i}, P_{S'_i}). \tag{33}$$

If the two arguments are true, by summing up (32) and (33), we can obtain the desired result (31).

We prove the two above Arguments.

**Argument 1.** Let

$$M_n := \sum_{t=1}^{n} \log \frac{P_{S_{I_t}}(X_{I_t,t})}{P_{S'_{I_t}}(X_{I_t,t})} - \mathrm{KL}(P_{S_{I_t}}, P_{S'_{I_t}}), \quad n = 1, 2, \ldots$$

and $M_0 := 0$.

We **firstly** prove that $(M_n)_{n=0}^\infty$ is a martingale w.r.t. $(\mathcal{H}_n)_{n=0}^\infty$: (1) $\mathbb{E}_\nu[|M_n|] < \infty$, and (2) $\mathbb{E}_\nu[M_{n+1} \mid \mathcal{H}_n] = M_n$.

(1) $\mathbb{E}_\nu[|M_n|] < \infty$. According to Assumption D.4, for any $i \in [K]$, $\mathrm{KL}(P_{S_i}, P_{S'_i}) < \infty$, this indicates there exists $c \in \mathbb{R}$ such that

$$\sum_{x=1}^{L+1} P_{S_i}(x) \Big| \log \frac{P_{S_i}(x)}{P_{S'_i}(x)} \Big| < c < \infty, \quad \forall i \in [K]. \tag{34}$$

This indicates

$$\mathbb{E}_\nu[|M_n|] \leq \sum_{t=1}^n \mathbb{E}_\nu \Big[ \Big| \log \frac{P_{S_{I_t}}(X_{I_t,t})}{P_{S'_{I_t}}(X_{I_t,t})} \Big| + \Big| \mathrm{KL}(P_{S_{I_t}}, P_{S'_{I_t}}) \Big| \Big] < \infty. \tag{35}$$

(2) $\mathbb{E}_\nu[M_{n+1} \mid \mathcal{H}_n] = M_n$. By adopting the tower property.

$$\mathbb{E}_\nu[M_{n+1} \mid \mathcal{H}_n] = M_n + \mathbb{E}_\nu \Big[ \mathbb{E}_\nu \Big[ \log \frac{P_{S_{I_{n+1}}}(Y_{I_{n+1},n+1})}{P_{S'_{I_{n+1}}}(Y_{I_{n+1},n+1})} - \mathrm{KL}(P_{S_{I_{n+1}}}, P_{S'_{I_{n+1}}}) \Big| \mathcal{H}_n, I_{n+1} \Big] \Big| \mathcal{H}_n \Big]$$

$$= M_n. \tag{36}$$

From (35) and (36), $(M_n)_{n \in \mathbb{N}}$ is a martingale w.r.t. $(\mathcal{H}_n)_{n \in \mathbb{N}}$.

**Additionally**, we will adopt Doob's Optional Stopping Theorem (Lemma F.1) on $M_{\mathrm{ST}(\texttt{ALG})}$. The prerequisites are verified as follows:

(1) $\mathbb{E}[\mathrm{ST}(\texttt{ALG})] < \infty$: By Assumption 4.2, $\mathrm{ST}(\texttt{ALG})$ is a stopping time w.r.t. $(\mathcal{H}_n)_{n \in \mathbb{N}}$ with $\mathbb{E}[\mathrm{ST}(\texttt{ALG})] \leq \mathrm{len}(\mathrm{pt}_{\tau_c} < \infty$.

(2) there exists $\bar{c} \in \mathbb{R}$, such that $\mathbb{E}[|M_{n+1} - M_n| \mid \mathcal{H}_n] \leq \bar{c}$ for any $n \leq \mathrm{ST}(\texttt{ALG})$: According to (34),

$$\mathbb{E}[|M_{n+1} - M_n| \mid \mathcal{H}_n] \leq \mathbb{E}_\nu \Big[ \Big| \log \frac{P_{S_{I_t}}(X_{I_t,t})}{P_{S'_{I_t}}(X_{I_t,t})} \Big| \Big| \mathcal{H}_n \Big] + \mathbb{E}_\nu \Big[ \mathrm{KL}(P_{S_{I_t}}, P_{S'_{I_t}}) \Big| \Big] \leq 2c < \infty.$$

Taking $\bar{c} = 2c$ finishes the verification.

Therefore, the prerequisites in Lemma F.1 (b) are satisfied. By invoking Lemma F.1, (32) is established.

**Argument 2.** Firstly, by (34),

$$\mathbb{E}_\nu \Big[ \Big| \sum_{t=1}^{\mathrm{ST}(\texttt{ALG})} \mathrm{KL}(P_{S_{I_t}}, P_{S'_{I_t}}) \Big| \Big] \leq \mathbb{E}_\nu \big[ \mathrm{ST}(\texttt{ALG}) \big] \cdot c < \infty.$$

So $\sum_{t=1}^{\mathrm{ST}(\texttt{ALG})} \mathrm{KL}(P_{S_{I_t}}, P_{S'_{I_t}})$ has finite expectation. Furthermore, it can be observed that

$$\mathbb{E}_\nu \Big[ \sum_{t=1}^{\mathrm{ST}(\texttt{ALG})} \mathrm{KL}(P_{S_{I_t}}, P_{S'_{I_t}}) \Big] = \mathbb{E}_\nu \Big[ \sum_{i=1}^K \sum_{t=1}^{\mathrm{ST}(\texttt{ALG})} \mathbb{1}\{I_t = i\} \mathrm{KL}(P_{S_i}, P_{S'_i}) \Big]$$

$$= \sum_{i=1}^K \mathbb{E}_\nu[n_{i,\mathrm{ST}(\texttt{ALG})}] \mathrm{KL}(P_{S_i}, P_{S'_i}).$$

Therefore, (33) is proved.

This concludes the proof of this divergence decomposition lemma. $\qquad\square$

### E.3. Proof of Proposition 4.5

**Proposition 4.5** (Tightness Result). *Let $\mathcal{S}_{\text{TGD}} = \{S : P_S \text{ satisfies (4)}\}$. Let $\{S_i\}_{i=1}^K \subset \mathcal{S}_{\text{TGD}}$ and $S_i$ satisfies (4) with $p_i$ (Line 5 in Algorithm 2), then*

$$\liminf_{m \to \infty} \frac{\text{Reg}(\texttt{ALG}, \text{pt}^m, \nu)}{\log(\text{len}(\text{pt}_{\tau_c}^m))} \geq \text{H}(\text{pt}, \nu) \cdot \frac{p_{i^*}(1 - p_{i^*}^L)}{(1 - p_{i^*})}.$$

*Therefore, the upper and lower bound match up absolute constants and a $\frac{L^2(1-p_{i^*})}{p_{i^*}(1-p_{i^*}^L)}$ factor. In particular, if $p_{i^*} \in \left(2^{-1/L}, 1\right)$, they match up to absolute constants and $L$.*

*Proof of Proposition 4.5.* Given any $S \in \mathcal{S}$ with parameter $p$,

$$\mu_S = \sum_{x=1}^{L+1} x \cdot P_S(x) = \sum_{x=1}^{L} x \cdot p^{x-1}(1 - p) + (L + 1) \cdot p^L = \frac{1 - p^{L+1}}{1 - p}.$$

Note that if $\mu_S \geq \mu_i$, we have $p > p_i$. Therefore,

$$\begin{aligned} \mu_S - \mu_i &= \frac{1 - p^{L+1}}{1 - p} - \frac{1 - p_i^{L+1}}{1 - p_i} = \frac{(p - p_i) + (p_i^{L+1} - p^{L+1} + p_i p^{L+1} - p p_i^{L+1})}{(1 - p)(1 - p_i)} \\ &\geq \frac{(p - p_i) + (p_i^{L+1} - p^{L+1})}{(1 - p)(1 - p_i)} \geq \frac{(p - p_i)(1 - p^L)}{(1 - p)(1 - p_i)}. \end{aligned} \tag{37}$$

In addition, for any $i \in [K]$,

$$\text{KL}(P_{S_i}, P_S) = \sum_{x=1}^{L+1} P_{S_i}(x) \cdot \log \frac{P_{S_i}(x)}{P_S(x)} = \frac{p_i - p_i^{L+1}}{1 - p_i} \cdot \log \frac{p_i}{p} + (1 - p_i^L) \log \frac{1 - p_i}{1 - p}.$$

By utilizing $\log x \leq x - 1$ for $x > 0$,

$$\text{KL}(P_{S_i}, P_S) \leq \frac{p_i - p_i^{L+1}}{1 - p_i} \cdot \frac{p_i - p}{p} + (1 - p_i^L) \frac{p - p_i}{1 - p} = \frac{(1 - p_i^L)(p_i - p)^2}{p(1 - p_i)(1 - p)} \tag{38}$$

Combining (38) and (37),

$$\text{KL}(P_{S_i}, P_S) \leq \frac{(\mu_S - \mu_i)^2(1 - p_i)(1 - p)(1 - p_i^L)}{p(1 - p^L)^2} \leq \frac{(\mu_S - \mu_i)^2}{p(1 - p^L)/(1 - p)}.$$

According to the definition of $\text{kl}_i$ in Theorem 4.4,

$$\text{kl}_i \leq \frac{(\mu_{i^*} - \mu_i)^2}{p_{i^*}(1 - p_{i^*}^L)/(1 - p_{i^*})} = \frac{\Delta_i^2}{p_{i^*}(1 - p_{i^*}^L)/(1 - p_{i^*})}.$$

Thus, the regret is lower bounded by

$$\liminf_{m \to \infty} \frac{\text{Reg}(\texttt{ALG}, \text{pt}^m, \nu)}{\log(\text{len}(\text{pt}_{\tau_c}^m))} \geq \sum_{i \neq i^*} \frac{1}{\mu_{i^*} \Delta_i} \cdot \frac{p_{i^*}(1 - p_{i^*}^L)}{(1 - p_{i^*})}.$$

Furthermore, if $p_{i^*} \in \left(2^{-1/L}, 1\right)$,

$$\liminf_{m \to \infty} \frac{\text{Reg}(\texttt{ALG}, \text{pt}^m, \nu)}{\log(\text{len}(\text{pt}_{\tau_c}^m))} \geq \sum_{i \neq i^*} \frac{L/2}{\mu_{i^*} \Delta_i}.$$

$\square$

## F. Supporting Lemmas

**Lemma F.1** (Doob's optional stopping, Theorem 3.8 in Lattimore & Szepesvári (2020)). *Let $\mathbb{F} = (\mathcal{F}_t)_{t \in \mathbb{N}}$ be a filtration and $(X_t)_{t \in \mathbb{N}}$ be an $\mathbb{F}$-adapted martingale and $\tau$ an $\mathbb{F}$-stopping time such that at least one of the following holds:*

*(a) There exists an $n \in \mathbb{N}$ such that $\mathbb{P}[\tau > n] = 0$.*

*(b) $\mathbb{E}[\tau] < \infty$, and there exits a constant $c \in \mathbb{R}$ such that for all $t \in \mathbb{N}$, $\mathbb{E}[|X_{t+1} - X_t| \mid \mathcal{F}_t] \leq c$ almost surely on the event that $\tau > t$.*

*(c) There exists a constant $c$ such that $|X_{t \wedge \tau}| \leq c$ almost surely for all $t \in \mathbb{N}$.*

*Then $X_\tau$ is almost surely well-defined, and $\mathbb{E}[X_\tau] = \mathbb{E}[X_0]$. Furthermore, when $(X_t)$ is a super/sub-martingale rather than a martingale, then equality is replaced with less/greater-than, respectively.*

**Lemma F.2** (Chernoff-Hoeffding bound, Fact 1 in Auer et al. (2002b)). *Let $X_1, \ldots, X_n$ be random variables with common range $[0, 1]$ and $\mathbb{E}[X_n \mid X_1, \ldots, X_{n-1}] = \mu$. Let $S_n = X_1 + \ldots + X_n$. Then for all $a \geq 0$,*

$$\mathbb{P}[S_n \geq n\mu + a] \leq \exp\left(-\frac{2a^2}{n}\right) \quad \text{and} \quad \mathbb{P}[S_n \geq n\mu - a] \leq \exp\left(-\frac{2a^2}{n}\right)$$

**Lemma F.3** (Confidence Intervals, Lemma 6 in Abbasi-yadkori et al. (2011)). *Assuming that the noise $\eta_t$ is conditionally 1-sub-Gaussian. With probability at least $1 - \delta$,*

$$\forall i \in \{1, 2, \ldots, K\}, \; \forall t \geq 0, \quad |\hat{\mu}_{i,t} - \mu_i| \leq \sqrt{\frac{(1 + n_{i,t})}{n_{i,t}^2}\left(1 + 2\log\left(\frac{K(1 + n_{i,t})^{1/2}}{\delta}\right)\right)}.$$

**Lemma F.4** (Lemma 8 in Antos et al. (2010)). *Let $a > 0$. For any $t \geq (2/a)[\log(1/a) - b]^+, at + b > \log t$.*

**Lemma F.5** (Exercise 3.7 in Lattimore & Szepesvári (2020)). *Let $\mathbb{F} = (F_t)_{t \in \mathbb{N}}$ be a filtration, and $\tau$ be a stopping time with respect to $\mathbb{F}$. Then $\mathcal{F}_\tau$ is a $\sigma$-algebra.*

**Lemma F.6** (Bretagnolle-Huber inequality, Theorem 14.2 in Lattimore & Szepesvári (2020)). *Let $P$ and $Q$ be probability measures on the same measurable space $(\Omega, \mathcal{F})$, and let $A \in \mathcal{F}$ be an arbitrary event. Then*

$$P(A) + Q(A^c) \geq \frac{1}{2}\exp\big(\mathrm{KL}(P, Q)\big),$$

*where $A^c = \Omega \setminus A$ is the complement of $A$.*

**Lemma F.7** (Pinsker's inequality, Equation (14.12) in Lattimore & Szepesvári (2020)). *For measures $P$ and $Q$ on the same probability space $(\Omega, \mathcal{F})$ that*

$$d_{TV}(P, Q) \leq \sqrt{\frac{1}{2}\mathrm{KL}(P, Q)}.$$

**Lemma F.8** (Theorem 26.13 in Lattimore & Szepesvári (2020)). *Let $\eta > 0$ and $f$ be Legendre and twice differentiable with positive definite Hessian in $A = \mathrm{int}(\mathrm{dom}(f))$. Then for all $x, y \in A$, there exists a $z \in [x, y] = \{(1-\alpha)x + \alpha y : \alpha \in [0, 1]\}$ such that*

$$\langle x - y, u \rangle - \frac{D_f(x, y)}{\eta} \leq \frac{\eta}{2}\|u\|_{\nabla^2 f(z)^{-1}}^2.$$

## G. Additional Experimental Results

### G.1. Additional Experimental Details

For the experiments stated in Section 6.1, we report the memory utilization in this section. As Ealge-2 (Li et al., 2024b) is one of the best speculative decoding methods, we adopt it as the baseline. The Normalized Memory (NM) and Normalized Memory Bandwidth (NMB) are presented in Table G.1. The result shows that the proposed methods do not incur additional memory consumption compared to the baseline method.

We further remark that this result is achieved by our superior algorithm design, where several *non-parametric* models (PLD, REST, Suffix Tree) to enhance a parametric SOTA model (Eagle-2). Specifically,

*Table 2.* The memory and memory bandwidth utilized by our method. As Eagle-2 is one of the best SD methods, we adopt it as the baseline to normalize the results of other methods. NM=Normalized Memory and NMB=Normalized Memory Bandwidth.

| Methods | Spec Bench | | Alpaca | | Code Editor | | Debug Bench | |
|---|---|---|---|---|---|---|---|---|
| | NM | NMB | NM | NMB | NM | NMB | NM | NMB |
| *LLaMA3-8B-Instruct* | | | | | | | | |
| Eagle-2 | 1.0000 | 1.0000 | 1.0000 | 1.0000 | 1.0000 | 1.0000 | 1.0000 | 1.0000 |
| EXP3SPEC | 0.9981 | 1.0171 | 0.9950 | 1.0170 | 1.0200 | 0.9980 | 1.0100 | 0.9960 |
| UCBSPEC | 1.0059 | 1.0093 | 1.0130 | 1.0090 | 0.9990 | 0.9820 | 1.0150 | 1.0020 |
| *Qwen2-7B-Instruct* | | | | | | | | |
| Eagle-2 | 1.0000 | 1.0000 | 1.0000 | 1.0000 | 1.0000 | 1.0000 | 1.0000 | 1.0000 |
| EXP3SPEC | 1.0043 | 1.0095 | 1.0050 | 0.9980 | 1.0400 | 0.9850 | 0.9890 | 0.9960 |
| UCBSPEC | 0.9929 | 1.0036 | 1.0080 | 0.9900 | 1.0270 | 0.9930 | 1.0320 | 0.9950 |

- "*Non-parametric*" means that these methods do not have any parameters in GPU, and directly predict the future tokens based on the past tokens according to the data structures like Trie Tree, which are python objects and stored in CPU RAM. All these show that the storage of the draft models will not increase the GPU memory. Our model only requires approximately an additional 100MB of CPU RAM. Since CPU memory is typically much larger (1TB in our server) and cheaper than GPU memory (40 GB in our server), this cost is negligible.

- All the draft models share the same verifier model, which is the target model (LLaMA3-8B-Instruct (Dubey et al., 2024) and Qwen2-7B-Instruct (Yang et al., 2024) in our experiments). So that the storage of the verifier does not increase the GPU memory.

- The reduction in memory usage comes from the fact that non-parametric models require fewer verification tokens (e.g., 40 for Suffix Tree) compared to the baseline Eagle-2 (e.g., 64). As a result, when invoking these models, a slight decrease in activation memory usage may be observed. Additionally, slight differences in GPU memory may be observed, arising randomly from the short-lived activation tensors rather than from the method itself.

We note that the size of SpecBench is not large enough, i.e., the number of arms pulls is not large, to derive a statistically sound result. We enable Mixture-of-Agent (Wang et al., 2024) on the prompts whose responses are shorter than 100 tokens to increase the number of arm pulls.

### G.2. Experiments on Larger Models

In addition to the two models in the main paper, we conduct an addtional set of experiment on a larger target model, namely LLaMA-2-13B (Touvron et al., 2023). As Table 1 indicates Eagle-2 (Li et al., 2024b) is one of the best speculative decoding methods, we adopt it as the baseline. The other setups are the same as the ones in Section 6.1.

From the result reported in Table G.2, the proposed methods, UCBSPEC and EXP3SPEC, demonstrate their efficacy on larger models.

### G.3. Experiments on Different Hardwares

In the main paper, the experiments are conducted on a single A100 GPU. In this section, we conduct an additional set of experiment on GeForce RTX 4090. We adopt Eagle-2 (Li et al., 2024b) as the baseline and Spec Bench (Xia et al., 2024) as the benchmark. The result is presented in Table G.3. We observe a similar trend as the result presented in Table 1. The proposed method remains useful with a different hardware setup.

*Table 3.* Empirical Comparison between the proposed algorithms and Eagle-2 (Li et al., 2024b) with LLaMA-2-13B as the target model, measured by Mean Accepted Tokens (MAT) (↑) and Tokens/s (↑). The best result is highlighted in **bold**, while the second best result is underlined. The proposed algorithms remain effective on larger models.

| Methods | Spec Bench | | Alpaca | | Code Editor | | Debug Bench | |
|---|---|---|---|---|---|---|---|---|
| | MAT(↑) | Tokens/s(↑) | MAT(↑) | Tokens/s(↑) | MAT(↑) | Tokens/s(↑) | MAT(↑) | Tokens/s(↑) |
| *LLaMA-2-13B* | | | | | | | | |
| Eagle-2 | 4.35 | 91.94 | 4.32 | 96.59 | 5.19 | 107.57 | 5.16 | 108.45 |
| EXP3SPEC | 4.05 | 95.52 | 4.32 | 99.64 | 5.22 | **115.65** | 5.03 | 116.65 |
| UCBSPEC | **4.43** | **97.16** | **4.36** | **102.29** | **5.27** | 113.97 | **5.27** | **118.67** |

*Table 4.* Empirical comparison between Eagle-2 and the proposed algorithms on GeForce RTX 4090. We observe a similar trend as the result presented in Table 1.

| Methods | Spec Bench | |
|---|---|---|
| | MAT | Tokens/s |
| *LLaMA3-8B-Instruct* | | |
| Eagle-2 | 4.14 | 97.01 |
| EXP3SPEC | 3.95 | 102.24 |
| UCBSPEC | **4.16** | **107.38** |
| *Qwen2-7B-Instruct* | | |
| Eagle-2 | 3.65 | 94.16 |
| EXP3SPEC | 3.96 | 111.74 |
| UCBSPEC | **4.17** | **112.21** |

