# OpenReview forum: "BanditSpec: Adaptive Speculative Decoding via Bandit Algorithms"
_ICML.cc/2025/Conference — ICML 2025 poster_

### Official Review · Reviewer_XwVz · 2025-03-14

**Overall Recommendation:** 2

**Summary:**

This paper introduces BANDITSPEC, which adaptively selects configurations for speculative decoding to improve inference speed. Unlike previous approaches that use fixed speculative decoding configurations regardless of context, BANDITSPEC formulates hyperparameter selection as a Multi-Armed Bandit problem, enabling dynamic adaptation to different inputs. The authors develop two key algorithms: UCBSPEC for stochastic environments and EXP3SPEC for adversarial settings, both with theoretical stopping time regret guarantees. They demonstrate BANDITSPEC's performance through extensive experimentation with LLaMA3 and Qwen2 models, showing that adaptive configuration selection outperforms existing fixed methods, approaching the performance of oracle best configurations. The framework proves particularly effective in real-world LLM serving scenarios with diverse input prompts, establishing a theoretically sound approach to minimize speculative decoding latency.

**Claims And Evidence:**

The paper provides solid theoretical guarantees for their claims through rigorous mathematical analysis, including lower bounds for the regret of their algorithms (UCBSPEC and EXP3SPEC). This theoretical foundation is a strength of the work. However, the experimental evidence has several limitations:

1. Incomplete competitor comparison: The paper lacks comparisons with direct competitors in adaptive speculative decoding, such as SpecDec++. This omission makes it difficult to fully evaluate BANDITSPEC's relative performance against state-of-the-art adaptive approaches.
2. Resource utilization metrics: While the authors demonstrate speedup compared to several baseline algorithms (vanilla, PLD, Rest, Suffix Tree, Eagle-2), they do not provide crucial metrics on resource utilization - specifically memory consumption and memory bandwidth utilization. This is particularly important since BANDITSPEC utilizes multiple speculative decoding algorithms, which likely has implications for resource overhead.
3. Limited hardware scenarios: Experiments are conducted only on a single A100 GPU with batch size 1, with limited exploration of more diverse computational environments that would be encountered in production settings. Speculative decoding is more suitable for edge applications and not as effective on data-center applications with high batch-size.

**Essential References Not Discussed:**

As mentioned, some direct competitors such as SpecDec++ was not discussed nor compared with.

**Experimental Designs Or Analyses:**

The concerns regarding the experimental designs are mentioned above.

**Methods And Evaluation Criteria:**

Some of the concerns regarding the evaluation criteria mentioned above. In addition, some of the numbers in the experiments do not match with the source numbers. For instance, the Llama 3.1 8B Spec Dec using Eagle 2 numbers in the respective paper is higher than the one mentioned in the paper. It might be due to different hyperparameters.

**Other Comments Or Suggestions:**

NA

**Other Strengths And Weaknesses:**

Mentioned above.

**Questions For Authors:**

Mentioned above.

**Relation To Broader Scientific Literature:**

This proposal can be beneficial for practical use cases of speculative decoding in memory-bound settings.

**Theoretical Claims:**

I examined the key theoretical proofs in the paper, particularly those related to the regret bounds in Theorems 4.3 and 5.3, which establish guarantees for UCBSPEC and EXP3SPEC respectively.
The proofs appear technically sound, with appropriate application of martingale theory and self-normalized concentration bounds to handle the unique challenges of the stopping time regret minimization problem.

---

> ### Author Rebuttal · Authors · 2025-04-01
>
> We thank the reviewer for the detailed reply and acknowledge the soundness of our theoretical results. We answer the questions regarding the experiments as follows:
>
> >**Q1**: Incomplete competitor comparison with adaptive speculative decoding algorithms like SpecDec++
>
> Thank the reviewer for the pointing this good work.
>
> - Firstly, we highlight that our proposed method is **training-free** which can be deployed easily along with **existing off-the-shelf methods**.
> In contrast, SpecDec++ focuses on **training** of an acceptance prediction head. Currently, SpecDec++ is only available when using LLaMA-2-Chat-7B as the draft model and LLaMA-2-Chat-70B as the target model (bfloat 16). It remains unclear how to integrate SpecDec++ with other potentially superior draft models/methods beyond LLaMA-2-Chat-7B. This lack of flexibility poses challenges for our implementation.
>
> - Secondly, the proposed BanditSpec framework considers the more general hyperparameter selection problem that goes beyond merely the speculation length. Therefore, it is "orthogonal" to SpecDec++ in the sense that any methods with (or without) SpecDec++ can also be candidates for the hyperparameter in our framework, e.g., {Eagle-2, LLaMA-2-Chat-7B} with SpecDec++ can also be regarded as arms (if they are available).
>
> - Thirdly, our ultimate goal is to devise algorithms that are competitive compared to the SOTA method, which is Eagle-2 (Li et al., 2024b). Given the currently available experimental results on the Alpaca dataset,  the speedup measured by the throughput (tokens/s) is as follows:
>
> | Methods                                   | Target Model        | Speedup                        |
> | --- | ------ | --- |
> | SpecDec++ with LLaMA-2-Chat-7B as drafter | LLaMA-2-Chat-70B    | 2.04 (Huang et al., 2024)                       |
> | Eagle-2                                   | LLaMA-2-Chat-70B    | 3.51 (Li et al., 2024b)               |
>
> Given the superior performance of Eagle-2, we adopt it as the backbone and baseline in our experiments.
>
> We will include the discussion about SpecDec++ in our revised version.
>
>
> >**Q2**: Resource utilization metrics
>
> Thank the reviewer for the advice! We highlight that the arm set for the model selection problem consists of one parametric model (Eagle-2) and several non-parametric models (PLD, Rest, Suffix Tree). These non-parametric models hardly consume the GPU resources. We measure the memory consumption and memory bandwidth utilization of our method. As Eagle-2 is one of the best SD method, we adopt it as the baseline to normalize the results of other methods. The result is accessible via this anonymous link [Table_Memory_Utilization](https://ibb.co/TBDQ0wZ2).
>
> In our experiments, slight differences in GPU memory usage were observed, arising randomly from the short-lived activation tensors rather than from the method itself.
>
> Although speculative decoding increases the reuse of I/O operations through parallel decoding, it does **not directly** affect memory bandwidth utilization. This is because memory bandwidth measures how much data can be transferred per unit time. According to our experiments, the memory bandwidth utilization of our approach is about the same as EAGLE-2.
>
> We will incorporate these discussions in the revised version.
>
> >**Q3**: Limited hardware scenarios
>
> Thank the reviewer for the question.
> We clarify that we do indeed investigate different computational setups. In Experiment 1, the batch size is to be 1 in order to compare the performance between the proposed method and the existing ones. In Experiment 2, we model the real-life scenario with diverse inputs and various batch sizes (ranging from 1 to 50) across the sample indices. According to the results in Figure 3, the throughput improvement is greater than 1 in most cases. This indicates the application of speculative decoding is still beneficial under reasonably high batch sizes.
>
> Additionally, we conduct our experiments on GeForce RTX 4090, whose result is accessible via this anonymous link [Table_Empirical_Comparison_on_4090](https://ibb.co/DDhL7BJh). We observe a similar trend as the result presented in Table 1 of the manuscript. The proposed method remains useful under this setup.
>
>
> If the reviewer has any other suggestions on other hardware scenarios for us to investigate, we would be happy to conduct such experiments to improve our paper.
>
> **We hope our responses have addressed your concerns and would greatly appreciate your kind consideration in increasing your score.**

---

### Official Review · Reviewer_S6Tu · 2025-03-14

**Overall Recommendation:** 3

**Summary:**

The paper introduces BanditSpec, a training-free online learning framework to route prompts to suitable off-the-shelf specualtive decoding methods. The authors formulate the problem as a Multi-Armed Bandit (MAB) problem and propose two bandit-based algorithms, UCBSpec and EXP3Spec, to adaptively choose different draft models and speculation lengths.

The paper provides theoretical analysis, including upper bounds on the stopping time regret under both stochastic and adversarial reward settings, and demonstrates the effectiveness of the proposed algorithms through empirical experiments with LLaMA3 and Qwen2 models. The results show that the proposed algorithms achieve competitive performance compared to existing methods, with throughput close to the oracle best hyperparameter in simulated real-life LLM serving scenarios.

**Claims And Evidence:**

The claims made in the paper are generally supported by clear evidence (though I didn't have a chance to check the proof details). The authors provide a detailed theoretical analysis, including upper bounds on the stopping time regret, and demonstrate the effectiveness of their algorithms through numerical experiments.

**Essential References Not Discussed:**

N/A

**Experimental Designs Or Analyses:**

The experimental designs and analyses can benefit from a more detailed analysis of the following aspects:

1. Computational resource overheads: The proposed method requires loading all base model weights into the GPU, which may introduce significant computational overhead compared to baseline methods. The paper does not provide a detailed discussion of the inference overheads, such as memory usage, GPU utilization, or latency introduced by the bandit algorithm itself. This information is crucial for understanding the practical feasibility of the proposed method, especially in resource-constrained environments.

2. Generation quality: It'd be great to assess and compare the generation quality of the proposed method with baselines. It is unclear whether the proposed method guarantees output parity with standard autoregressive decoding. For example, does the adaptive selection of hyperparameters introduce any degradation in the quality of the generated text?

**Methods And Evaluation Criteria:**

The proposed method uses a bandit framework to adaptively select hyperparameters (e.g., base speculative decoding strategies and number of speculations), which makes sense but does not seem very practical as it requires loading all base models into the GPU, significantly increasing the computational resource overheads.

The evaluation criteria can be improved by adding the numbers of actual performance on downstream tasks, providing a comparison of the generation quality of the proposed method and compared baselines.

**Other Comments Or Suggestions:**

Please address the concerns mentioned in the above sections.

**Other Strengths And Weaknesses:**

Other  Strengths:

- The second experiment is conducted in simulated real-life scenarios with diverse input prompts (however, no comparison with baseline models in this setting)

Other Weaknesses:
- The paper lacks a detailed analysis of the computational overheads introduced by the proposed method, such as memory usage and GPU utilization.
- The evaluation does not include an assessment of generation quality, which is critical for understanding the practical utility of the method.

**Questions For Authors:**

1. The proposed method requires loading all base model weights into the GPU, which may introduce significant computational overhead. Could the authors provide a detailed analysis of the inference overheads, including memory usage, GPU utilization, and latency? How do these overheads compare to existing methods?

2. The paper does not evaluate the generation quality of the proposed method. Does the adaptive selection of hyperparameters introduce any degradation in the quality of the generated text? Could the authors provide an evaluation of generation quality measured by downstream task metrics such as accuracy? How does the proposed method perform compared to standard autoregressive decoding in terms of downstream task performance?

**Relation To Broader Scientific Literature:**

The paper builds on existing sepcualtive decoding methods work by introducing a badnit framework to route input prompts to different off-the-self methods.

**Theoretical Claims:**

The authors derive upper bounds on the stopping time regret for both UCBSpec and EXP3Spec under stochastic and adversarial settings, and the proofs appear to be technically sound. However, the theoretical analysis relies on assumptions that may not hold in practical scenarios, which limits the practical value of the results. Specifically,

- Assumption of Fixed Prompts: The theoretical analysis assumes that the input prompts are fixed, which is rarely the case in real-world applications. In practice, prompts are highly diverse and dynamic, and the performance of speculative decoding methods can vary significantly depending on the input. This raises questions about how well the theoretical bounds translate to real-world decoding latency improvements.

- Stationary Mean Acceptance Length: The stochastic setting assumes that the mean number of accepted tokens for each hyperparameter is stationary (Assumption 4.1). This assumption may not hold in practice, as the acceptance rate of speculative tokens can vary depending on the context and the specific input prompt. This limits the applicability of the theoretical results to real-world scenarios where the acceptance rate is non-stationary.

- Adversarial Setting: While the adversarial setting relaxes the stationarity assumption, it still assumes that the number of accepted tokens is fixed by the environment before the algorithm starts (Assumption 5.1). This is also an unrealistic assumption in practice, as the acceptance rate can depend on the interaction between the draft model, the target model, and the input prompt.

I understand the intuition of such theoretical analysis is to provide insights into the behavior of the proposed algorithms under idealized conditions, but its practical implication is limited by such assumptions -- For example, a lower regret bound under such assumptions does not necessarily guarantee lower decoding latency in practical settings.

---

> ### Author Rebuttal · Authors · 2025-04-01
>
> We thank the reviewer for the detailed reading and feedback.
>
> >**Q1**: Computational overheads introduced by the proposed method, such as memory usage and latency.
>
>
> Thank the reviewer for the advice! For the memory and memory bandwidth usage, please kindly refer to this anonymous link [Table_Memory_Utilization](https://ibb.co/TBDQ0wZ2).
> For latency introduced by the bandit algorithm, we clarify that the latency (token/s) has already taken this factor into account. According to Table 1, the speedup of the proposed method is $\times 2.96$ with repsect to the vanilla SD method and is $\times 1.08$  with repsect to the SOTA, namely Ealge-2. In conclusion, the benefits of BanditSpec come at a negligible cost.
>
> >**Q2**: Generalization quality of the proposed method.
>
> In the speculative decoding literature, it is theoretically guaranteed that the distribution of the generated sequence is the **same** as that of the target model, which means the quality of the output is **maintained** and **lossless acceleration** is achieved.(see e.g., Leviathan et al., 2023, Chen et al., 2023, Yin et al., 2024). Therefore, the quality metric is often omitted in the experiments and more emphasis is put on the acceleration and latency metrics (see Leviathan et al., 2023, Chen et al., 2023, Yin et al., 2024, Li et al., 2024b, etc.).
>
> As we have also provided the same theoretical guarantees for the quality of the generated tokens from the BanditSpec framework in Proposition 1, the quality metric is omitted, following the convention in the community (see the above references).
>
> >**Q3**: Assumption of Fixed Prompts.
>
> We would like to clarify that the theoretical guarantees are derived to bound the latency given **any** prompt, as explained in Line 201 "Interpretation of the Desired Result". We completely agree that prompts are diverse and performance of the speculative decoding (SD) method can be significantly influenced by the input prompts. This is indeed why we propose the BanditSpec framework, where given an input prompt, BanditSpec gradually learns the best SD method for this specific input prompt as the decoding process proceeds. Observe that our objective in equation (1) is $\text{Reg}(\text{ALG}, \text{pt}, \nu)$, which depends explicitly on the prompt $\text{pt}$. We show that the additional SD rounds (the stopping time regret) is sublinear in the SD rounds required by the best SD method, indicating that the best SD method is adopted "most of the time" under BanditSpec. This perfectly aligns with real-world scenarios. We will further highlight this setup in the revised manuscript.
>
> >**Q4**: Stationary Mean Acceptance Length Assumption and its applicability to real-world scenarios.
>
> We understand that in real-world scenarios, there can be many factors that influence the acceptance rate, making it non-stationary or even adversarial. This is also why we relax the i.i.d. assumption that is commonly used in the standard stochastic Multi-Armed Bandits (MAB) and we do allow the acceptance rate to be dependent on the input prompts and generated tokens. In particular, the distribution of the acceptance rate can also change with the only constraint on its mean under our assumption. Additionally, our formulation under the stationary mean assumption paves the way for the application of more generalized setups, like contextual bandits and non-stationary bandits which can lead to future research.
>
> On the experimental side, the experimental results in Table 1 show that the proposed method exhibits competitive empirical performance compared to the current methods, including Eagle-2, the SOTA in SD, which strongly corroborates the validity of the assumptions. Additionally, the applicability of our theoretical results has also been empirically verified by the experimental results.
>
> >**Q5**: Adversarial Setting.
>
> We would like to clarify that we derive the results for the adversarial setting under the **greedy decoding** strategy. Given a draft model, a target model and an input prompt, under the greedy decoding strategy, the output tokens are **fixed** but **unknown** at the beginning of the algorithm. This is modeled precisely by our adversarial MAB setup.
>
> Furthermore, we include the adversarial setting in our paper as a means to compare it to the stochastic setting. Prior to our work, it was a prior unclear how to use MAB to improve SD. Should one employ a stochastic, adversarial or even more generalized model? We consider a range of such MAB models and do a comparison among them to provide the community with a guide on which MAB model is best suited to the SD problem.
> As the empirical performance of UCBSpec is better than EXP3Spec, it implies that real-life scenario tends to be benign and may be more aligned with the stationary mean assumption. We will highlight this observation in the revised version.
>
> **We hope our responses have addressed your concerns and would greatly appreciate your kind consideration in increasing your score.**

---

> > ### Comment · Reviewer_S6Tu · 2025-04-05
> >
> > Thank you for the responses and clarifications. However, I’m even more confused by the results shown in [Table_Memory_Utilization](https://ibb.co/TBDQ0wZ2), which suggest that the proposed method requires less memory usage compared to the baseline. This seems counterintuitive, since the UCBSpec/EXP3Spec needs to load all candidate methods (i.e., the corresponding draft and verifier models) into the GPU so that the UCBSpec/EXP3Spec can route the request to the best candidate. This ensemble of multiple methods is supposed to use more memory than simply running a single method. Yet, the results in Table_Memory_Utilization suggest otherwise. Could you clarify this?
> >
> > Additionally, while I appreciate the author's efforts in presenting theoretical analysis, it’s still unclear to me in what real-world application the assumption—that the mean of the distribution of acceptance rates is fixed—would hold. Could you provide a practical example or scenario where this assumption is reasonable?

---

> > > ### Author Response · Authors · 2025-04-05
> > >
> > > Thanks for the insightful comment.
> > >
> > > > **The GPU Memory Usage of Our Methods**
> > >
> > > Thanks for this insightful question. This is in fact an advantage of our algorithmic design, where we use several **non-parametric models (PLD, REST, Suffix Tree)** to enhance a parametric SOTA model(EAGLE).
> > >
> > > * Here the word ``non-parametric'' means that these methods **do not have any parameters in GPU**, and directly predict the future tokens based on the past tokens according to the **data structures** like Trie Tree, **which are python objects and stored in CPU RAM**. All these show that the storage of the draft models will not increase the GPU memory. Our model only requires approximately an additional 100MB of CPU RAM. Since CPU memory is typically much larger (1TB in our server) and cheaper than GPU memory (40 GB in our server), this cost is negligible.
> > >
> > > * In addition, we note that all the draft models share **the same verifier model**, which is the target model (Llama 3 and Qwen 2 in our experiments). So that the storage of the verifier does not increase the GPU memory.
> > >
> > >
> > > The reduction in memory usage comes from the fact that non-parametric models require fewer verification tokens (e.g., 40 for Suffix Tree) compared to the baseline EAGLE (e.g., 64). As a result, when invoking these models, a slight decrease in activation memory usage may be observed.
> > >
> > >
> > > > **Stationary Mean Values Assumption**
> > >
> > > Thank a lot for this question. We would like to further explain the applicability of Assumption 4.1.
> > >
> > > * We note that Assumption 4.1 and Theorem 4.3 hold in a **prompt-wise** sense. It means that Assumption 4.1 admits that the mean acceptance rates are different for different prompts, and it only requires the stationarity for each prompt.
> > >
> > > * We would like to provide some reasons for the stationarity of both parametric and non-parametric models. The parametric model is trained via the next-token prediction method. Thus, it treats the prediction of all the tokens in a **symmetric** way. Intuitively, such symmetry implies the stationarity in the average sense. The non-parametric models all use the past information for prediction. For example, in the code modification task, where models are called to modify the bugs in a given code, non-parametric models can predict the tokens **in a very stable way**, since the past information is very useful for the prediction.
> > >
> > > * We note that our methods are designed based on this assumption. The efficacy of it in the real-world setting across different models and datasets also partially verifies this assumption.
> > >
> > > We also note that there are several ways to generalize this assumption. For example, we can generalize it to the block-wise stationarity, i.e., this assumption holds in some continuous decoding steps. However, **the theoretical analysis and practical implementation of the generalization will be based on our theoretical techniques, especially the regret decomposition, and our codebase**. We leave them for future work.
> > > ___
> > >
> > > **We hope our responses have addressed your concerns and would greatly appreciate your kind consideration in increasing your score.**

---

### Official Review · Reviewer_pdNK · 2025-03-18

**Overall Recommendation:** 3

**Summary:**

This paper proposes a training-free online learning framework to adaptively choose the configuration of the hyperparameters for speculative decoding as text is being generated. Specifically, this paper first formulates this hyperparameter selection problem as a Multi-Armed Bandit problem, and proposes two bandit-based hyperparameter selection algorithms to adaptively select configurations for speculative decoding. Experiments with LLaMA3 and Qwen2 demonstrate that the proposed method is effective compared to existing methods.

## update after rebuttal

Thanks for the authors' detailed rebuttal. I will maintain my positive score.

**Claims And Evidence:**

Yes.

**Essential References Not Discussed:**

No.

**Experimental Designs Or Analyses:**

1.	(Strengths) Experiments demonstrate the proposed method outperforms existing methods in terms of inference latency.

2.	(Weaknesses) The authors use LLaMA3-8B-Instruct and Qwen2- 7B-Instruct as the target models. However, it would be more convincing to evaluate the proposed method on larger models.

**Methods And Evaluation Criteria:**

**Method**

1.	(Strengths) The proposed bandit-based online hyperparameter configuration method for speculative decoding is interesting and practical in real applications.

2.	(Strengths) The authors shown that the regret performance of the proposed method is optimal up to universal constants by deriving an information-theoretic impossibility result.

3.	(Weaknesses) The authors propose to formulate the draft model selection in standard speculative decoding as a multi-armed bandit problem. However, it simplifies the correlation between different decoding steps, which can be unaligned with realistic decoding process. It would be more convincing if the authors could provide more supports for the reasonableness of formulating draft model selection across different decoding steps as a over-simplified multi-armed bandit problem.

**Evaluation Criteria**

1.	(Strengths) Experiments demonstrate the proposed method outperforms existing methods in terms of inference latency.

2.	(Weaknesses) The authors use LLaMA3-8B-Instruct and Qwen2- 7B-Instruct as the target models. However, it would be more convincing to evaluate the proposed method on larger models.

**Other Comments Or Suggestions:**

No

**Other Strengths And Weaknesses:**

Please see the above comments.

**Questions For Authors:**

Please see the above comments.

**Relation To Broader Scientific Literature:**

1.	The authors propose to formulate the draft model selection in speculative decoding as a multi-armed bandit problem, which is an interesting formulation.

2.	The authors propose to leverage two widely-used multi-armed bandit methods to adaptively select the draft models, and provide theoretical guarantees of the proposed method under mild assumptions.

3.	Experiments demonstrate that the proposed method outperforms existing methods.

**Theoretical Claims:**

Yes, the theoretical claims are correct. However, it would be more convincing to explain the reasonableness of the assumptions.

---

> ### Author Rebuttal · Authors · 2025-04-01
>
> We thank the reviewer for the detailed feedback and helpful suggestions.
>
> >**Q1**: It would be more convincing if the authors could provide more supports for the reasonableness of formulating draft model selection across different decoding steps as a over-simpliﬁed multi-armed bandit problem.
>
> We thank the reviewer for the question.
> We believe the reviewer thinks that it is an oversimplification to employ the **standard stochastic Multi-Armed Bandits (MAB)** to model the Speculative Decoding (SD) problem, because the rewards (the number of accepted tokens in our case) of an arm (hyperparameter configuration) in the vanilla MAB problem are i.i.d. and hence, cannot capture the correlation between different decoding steps.
>
> On the theoretical side, we clarify that our stationary mean assumption is strictly **weaker** than the i.i.d. assumption (this is discussed in Line 209 on the second column of page 4). In particular, the number of accepted tokens **can depend on the generated tokens**. Therefore, the assumption is aligned with real-world scenarios in which different decoding steps are correlated. Furthermore, the basic MAB model can be generalized to contextual bandits and non-stationary bandits. The proposed BanditSpec framework provides a basic template to apply these more general MAB setups to SD. Our formulations under the stationary/adversarial mean assumptions are just basic setups and we leave the more general/elaborate setups as future research (please refer to Appendix B for more details).
>
> On the experimental side, as our experimental results indicate (Table 1), the performance of UCBSpec significantly outperforms the SOTA in SD, namely, Eagle-2 (Li et al., 2024). This **corroborates** the stationary mean assumption in our formulation.
>
> >**Q2**: Evaluation of the proposed method on larger models.
>
> Thank the reviewer for the advice. We further conduct the experiment with LLaMA-2-13B (Touvron ea al., 2023) as the target model. As Eagle-2 is one of the best SD methods, we adopt it as the baseline. The result is as follows:
>
>
>
> | Methods | Spec Bench |  | Alpaca |  | Code Editor |  | Debug Bench |  |
> |:---:|:---:|:---:|:---:|:---:|:---:|:---:|:---:|:---:|
> |  | MAT | Tokens/s | MAT | Tokens/s | MAT | Tokens/s | MAT | Tokens/s |
> | Eagle-2 | 4.35 | 91.94 | 4.32 | 96.59 | 5.19 | 107.57 | 5.16 | 108.45 |
> | EXP3Spec | 4.05 | 95.52 | 4.32 | 99.64 | 5.22 | **115.65** | 5.03 | 116.65 |
> | UCBSpec | **4.43** | **97.16** | **4.36** | **102.29** | **5.27** | 113.97 | **5.27** | **118.67** |
>
>
> This indicates the proposed BanditSpec framework is useful on larger models. We will incorporate the results in the revised version.
>
> **We hope our responses have addressed your concerns and would greatly appreciate your kind consideration in increasing your score.**

---

> > ### Comment · Reviewer_pdNK · 2025-04-04
> >
> > Thanks for the authors' response. I will maintain my original score.

---

### Decision · Program_Chairs · 2025-05-01

**Decision:**

Accept (poster)

**Comment:**

This paper proposes a multi armed bandit based framework to adaptively choose the configuration of the hyperparameters for speculative decoding. The reviews mention the approach as convincing and appreciated the theoretical analysis (e.g., XwVz: “solid theoretical guarantees for their claims through rigorous mathematical analysis, including lower bounds for the regret of their algorithms”). They also mentioned the empirical analysis to be convincing of the value of the method. The latter is important since the theoretical analysis relied on assumptions that may not always hold, and in fact were mentioned as unrealistic.

A few concerns were raised by the reviewers, and were addressed during the rebuttal stage: (1) Impracticality due to memory overhead (S6Tu, “the proposed method requires loading all base model weights into the GPU, which may introduce significant computational overhead compared to baseline methods.”): The authors analyzed the memory overhead, showing it is negligible, and in CPU memory. (2) Experiments require more baselines, XwVz: “The paper lacks comparisons with direct competitors in adaptive speculative decoding, such as SpecDec++.”: Here, the authors mentioned that the mentioned baselines are compared with the Eagle-2 method that performed better, and this method was tested in the paper. (3) Additional experiments needed: More scenarios (mentioned by XwVz), additional LLMs (mentioned by pdNK). The authors provided experiments in the rebuttal.

Given these additional experiments and clarifications provided in the rebuttal, the remaining concerns are quite mild, making this paper sufficient in quality for ICML. Furthermore, the scope of the change does not seem very large to me and can be done towards a camera-ready version.